# Receptor Tyrosine Kinase KIT: Mutation-Induced Conformational Shift Promotes Alternative Allosteric Pockets

Julie Ledoux , Marina Botnari and Luba Tchertanov *

Centre Borelli, CNRS, ENS Paris-Saclay, Université Paris-Saclay, 4 Avenue des Sciences,
F-91190 Gif-sur-Yvette, France; julie.ledoux@ens-paris-saclay.fr (J.L.); marina.botnari@etu.u-paris.fr (M.B.)
* Correspondence: luba.tchertanov@ens-paris-saclay.fr

**Abstract:** Receptor tyrosine kinase (RTK) KIT is key regulator of cellular signalling, and its deregulation contributes to the development and progression of many serious diseases. Several mutations lead to the constitutive activation of the cytoplasmic domain of KIT, causing the aberrant intracellular signalling observed in malignant tumours. Elucidating the molecular basis of mutation-induced effects at the atomistic level is absolutely required. We report the first dynamic 3D model (DYNA-SOME) of the full-length cytoplasmic domain of the oncogenic mutant KIT$^{D816V}$ generated through unbiased long-timescale MD simulations under conditions mimicking the natural environment of KIT. The comparison of the structural and dynamical properties of multidomain KIT$^{D816V}$ with those of wild type KIT (KIT$^{WT}$) allowed us to evaluate the impact of the D816V mutation on each protein domain, including multifunctional well-ordered and intrinsically disordered (ID) regions. The two proteins were compared in terms of free energy landscape and intramolecular coupling. The increased intrinsic disorder and gain of coupling within each domain and between distant domains in KIT$^{D816V}$ demonstrate its inherent self-regulated constitutive activation. The search for pockets revealed novel allosteric pockets (POCKETOME) in each protein, KIT$^{D816V}$ and KIT$^{WT}$. These pockets open an avenue for the development of new highly selective allosteric modulators specific to KIT$^{D816V}$.

**Keywords:** receptor tyrosine kinase (RTK) KIT; oncogenic mutation D816V; molecular dynamics and folding; conformational plasticity; intrinsically disordered regions; IDRs; free energy landscape; intramolecular coupling; new allosteric pockets





## 1. Introduction

Receptor tyrosine kinases (RTKs) are canonical membrane proteins that control the signal transduction of extracellular signals to the nucleus through tightly coupled signalling cascades, altering the expression pattern of numerous genes [1–3]. RTK KIT, also known as the CD117 differentiation cluster, is an RTK family III consisting of 976 amino acids (aas). Its activity is regulated by a highly specific cytokine, the Stem-Cell Factor (SCF), acting as a messenger to initiate signal transduction. Stimulation by SCF in the extracellular medium enables KIT to recruit protein partners through the cytoplasmic domain. KIT initiates critical signalling pathways through multiple specific phosphotyrosines that bind downstream proteins containing Src homology (SH2) or phosphotyrosine-binding (PTB) domains and propagates the SCF-induced signal to nuclei [4–6].

The RTK KIT-activated signalling pathways control many important cellular processes such as proliferation, survival, migration, development, and functions of various cell types, including germ cells and immature haematopoietic cells [7]. Under physiological conditions, KIT gene expression, protein activity, and activated signalling processes are quantitatively and temporally perfectly controlled. Dysregulation of KIT activity underpins abnormal cell development leading to tumourigenesis [8,9]. In particular, constitutive activation may confer oncogenic properties upon normal cells and triggers RTK KIT-induced signalling independently of SCF stimulation [9,10]. KIT overexpression and gain-of-function mutations have been reported in

different types of cancer, such as gastrointestinal stromal tumours (GISTs) [11,12], small-cell lung carcinomas (SCLC) [13], acute myeloid leukaemias (AMLs) [14,15], melanomas [16], systemic mastocytosis [17,18], and others.

KIT's physiological functions are highly related to its molecular features—modular architecture, plasticity, and transmembrane location—providing tight cooperation of KIT's extracellular and cytoplasmic domains (ED and CD) through a single-pass helical linker, the transmembrane domain (TMD) (Figure 1a).

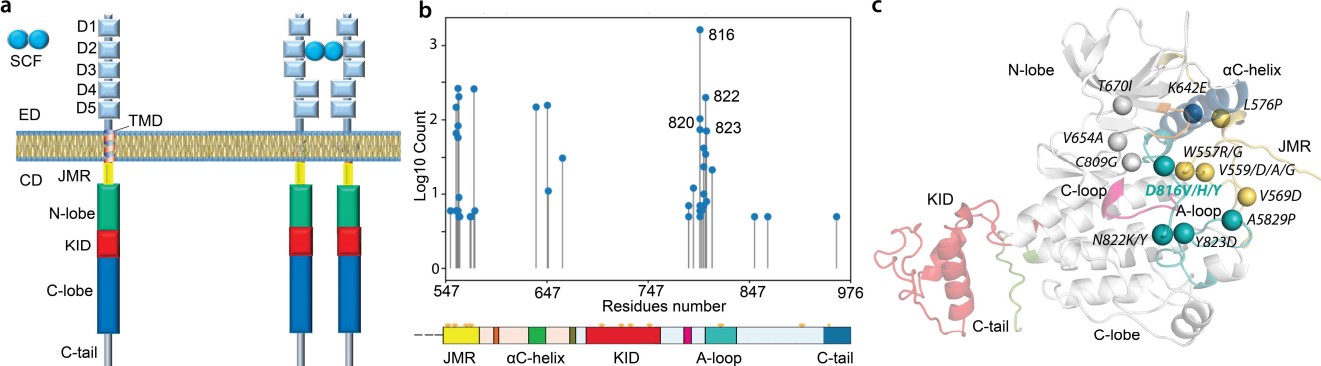

**Figure 1.** RTK KIT SCF-induced activation and its cytoplasmic missense mutations. (**a**) The modular architecture of RTK KIT in monomeric (SCF-unbound) and dimeric (SCF-bound) states, schematised on the left and right, respectively. The extracellular domain (ED) composed of five Ig-like domains (D1–D5) is linked by the transmembrane domain (TMD) to the cytoplasmic domain (CD) consisting of a juxtamembrane region (JMR), tyrosine kinase bi-lobe domain (TKD containing N- and C-lobes), kinase domain insert (KID), and C-terminal tail. SCF extracellular binding induces KIT dimerisation and activation. (**b**,**c**) The most studied somatic KIT CD mutations in cancer are presented by their occurrences (number of items mentioned in articles) (**left**) and their positions in the 3D structure (**right**).

SCF extracellular binding to KIT ED promotes the ligand-induced receptor dimerisation and CD activation. The departure of crucial functional regions—juxtamembrane region (JMR), catalytic αC-helix, and activation (A-) loop—from their autoinhibited positions (structural effects) and phosphorylation of specific tyrosine residues (biochemical/biophysical effects) lead to SCF-dependent signal transfer and controlled adaptor or scaffold proteins binding to KIT, followed by the initiation of intracellular signalling cascades [19,20].

Wild-type KIT (KIT$^{WT}$) activates various signalling pathways through multiple protein partners, either directly or indirectly linked to KIT, achieving sophisticated but finely regulated signalling cascades. Proteins such as Src family kinases (SFK) or phospholipase C (PLCγ) activate the Mitogen-Activated Protein Kinase (MAPK) signalling pathway through the RAS/RAF or JAK/STAT pathways, phosphatidylinositol 3-kinase (PI3K), its eponymous pathway, and, via RAC, the MAPK and c-Jun N-terminal kinase (JNK) pathways [5,21]. A large set of protein–protein complexes formed during the initiation of signalling cascades constitutes KIT's INTERACTOME.

The most common perturbations of KIT's signalling pathways result from KIT somatic and germline mutations. These gain-of-function mutations typically affect residues from regions involved in the inactive-to-active conformational transition and lead to the receptor's constitutive (SCF-binding independent) activation. Indeed, JMR, A-loop, and αC-helix, the main actors contributing to the KIT state transition (inactive-to-active), are the primary locations of gain-of-function mutations (Figure 1b,c). KIT mutation at position 816 in A-loop shows divergent substitutions (D816V, D816Y, D816F, *or* D816H) and is the most frequently studied for KIT-involved cancers. In particular, D816V is widely implicated in haematological malignancies, including mastocytosis and AMLs [12,17,22–24].

Receptor KIT$^{D816V}$ is constitutively active, and its aberrant activity has been explained by structural transformations in the JMR and A-loop, which lead to a disruption of the

coupling between these regions, and manifested as a disturbance of their communication pathways [25,26].

Recently, experimental evidence revealed that the KIT$^{D816V}$ mutant did not dimerise like KIT$^{WT}$ [10], and shows decreased tyrosine kinase domain stability and signalling amplification compared to the SCF-activated KIT$^{WT}$ [10,18,27]. Comparison of downstream signalling pathways activated by oncogenic KIT$^{D816V}$ versus KIT$^{WT}$, performed on qualitative and quantitative levels, exhibits the significant difference in signalling potential and alteration of downstream proteins contributing to KIT signalling pathways [28–33].

In terms of specific pathology-related involvement, KIT$^{D816V}$ was considered the primary disease driver in many illnesses and suggested as a crucial therapeutic target. Imatinib, developed by rational drug design, occupies a special place among the TK inhibitors and represents a multitarget drug (inhibits ABL, BCR-ABL, KIT, and PDGFR$\alpha$) widely used for the treatment of many tumours [34]. However, in vitro investigations of the imatinib efficacy revealed that although the drug effectively inhibits KIT$^{WT}$, it does not adequately inhibit KIT$^{D816V}$ [35,36]. The KIT$^{D816V}$ resistance to imatinib has prompted an in silico investigation explaining its mechanisms [37]. Numerous studies have led to the discovery of new highly effective tyrosine kinase inhibitors. For example, dasatinib, a potent multi-target drug targeting ABL, SRC, KIT, PDGFR, and other tyrosine kinases, demonstrates significant inhibitory activity against KIT$^{WT}$ and KIT$^{D816V}$ [38]. A 20-year effort has been rewarded with the development of two KIT-specific new molecules, ripretinib and avapritinib, selective for KIT and PDFFR$\alpha$, for the treatment of multidrug-resistant cancers [39,40]. Nevertheless, anticancer drugs targeting tyrosine kinases affect tumours but promote serious side effects, such as potential pulmonary or cardiovascular toxicities [41,42].

Therefore, a synergistic description of KIT$^{D816V}$ will provide prognostic information and open new avenues of research exploring alternative targeted therapeutic strategies, which is conceptually paramount.

In the present work, we examined the D816V mutation-induced effects on RTK KIT at the atomistic level using 3D model of the full-length cytoplasmic domain of KIT$^{D816V}$ bound to a TM helix embedded in a membrane. To distinguish the effects caused by D816V mutation, this model was explored by molecular dynamics (MD) simulations, and the generated data were carefully analysed and compared with that of KIT$^{WT}$ [43].

The multidomain modular KIT consists of a quasi-stable TKD crowned by four intrinsically disordered (ID) regions–JMR, KID, A-loop, and C-tail–each containing functional tyrosine residues. Those ID regions act either as a platform ground for the recruitment of signalling proteins (JMR, KID, and C-tail) or as main promoters of the KIT activation mechanisms (JMR and A-loop). Knowing such information, we focused on the following essential questions: (i) Does the D816V mutation affect kinase domain? (ii) Does the D816V carcinogenic mutation contribute to the order–disorder processes in IDRs of modular KIT? (iii) Which intrinsic disorder events—transient folding, conformational variability, or both—are the main factors contributing to the constitutively active state of KIT$^{D816V}$? (iv) Since a tight dynamical coupling between JMR, KID, and C-tail was observed in KIT$^{WT}$, we asked: is this coupling is retained in the KIT$^{D816V}$ mutant? (v) What biophysical factors are decisive in the formation of the constitutively active state of KIT$^{D816V}$? The answers to these questions will help establish the causes that lead to the dysregulation of KIT$^{D816V}$ signalling. Thus, clarifying the inherent dynamics of KIT (DYNASOME [44]), describing the ensemble of its pockets (POCKETOME [45]), and identifying KIT's interactions with all its protein partners (INTERACTOME [46]) could help develop highly effective KIT-specific inhibitors acting simultaneously on intramolecular targets and the interface between interacting proteins allo-network drugs [47].

The first step in such a study is to describe the KIT$^{D816V}$ DYNASOME in detail and compare it to the inactive KIT$^{WT}$. Such comparison will lead to the identification of D816V-induced effects on the biophysical properties of KIT, which is beneficial for interconnecting these effects with empirical gain-of-function data.

To the best of our knowledge, we report, for the first time, an exhaustive comparison of an inactive state of KIT$^{WT}$ and one of its oncogenic constitutively active mutants using the most complete and advanced 3D dynamical models.

## 2. Results

### 2.1. Data Generation and Proceeding

The KIT$^{D816V}$ 3D model (sequence I516-R946) was derived by homology modelling from the KIT$^{WT}$ full-length CD model with its transmembrane helix [43] (Figure S1) and studied by all-atom MD simulation in its natural environment (the protein was embedded into a membrane through the TM helix and submerged in water). Three independent 2-μs MD simulation replicates were generated to enhance conformational sampling and examine the consistency and completeness of KIT$^{D816V}$ conformations produced upon strictly identical conditions. The generated MD trajectories were analysed for a full-length construct and per domain/region using unique and concatenated trajectories. To avoid rigid body motions, KIT$^{D816V}$ trajectories were normalised by least-square fitting on the initial structure (t = 0 μs). For the comparative analysis of KIT$^{D816V}$ data with the previously published of KIT$^{WT}$ [43], an additional normalisation was performed by least-square fitting of all MD conformations to the same initial structure (KIT$^{WT}$, t = 0 μs).

### 2.2. General Characterisation of MD Trajectories

The root-mean-square deviations (RMSDs), computed for each KIT$^{D816V}$ MD conformation, display comparable profiles between replicated trajectories, demonstrating good reproducibility of the generated data (Figure S2). As observed in KIT$^{WT}$, KIT$^{D816V}$ RMSD values are mainly influenced by KID (up to 22 Å), JMR (up to 15 Å), and C-tail (up to 18 Å), while the TKD lobes show significantly smaller RMSD values (<4 Å). The profiles of the KIT$^{D816V}$ root-mean-square fluctuations (RMSFs) are comparable in the three MD trajectories, showing only differences in the highly fluctuating KID, as observed in KIT$^{WT}$. The RMSDs and RMSFs calculated after fitting on each domain show a systematic decrease in their values, indicating greater inter-domain effects in respect to intra-domain.

### 2.3. KIT Folding in Inactive (Wild Type) and Constitutively Active (Mutant) States

The KIT$^{D816V}$ TKD folding (2D structure) is generally well-conserved along the MD simulations (Figure 2). The average structural folding of TK domain N- and C-lobes corresponds well to the empirically (X-ray) characterised structures of inactive (PDP ID: 1T45) and active states of KIT$^{WT}$ (PDB ID: 1PKG) [48,49].

The fold of four intrinsically disordered regions (IDRs)—JMR, KID, A-loop, and C-tail—along the MD trajectories of KIT$^{D816V}$ show either partially conserved secondary structures or structures reversibly transitioning from folded to alternatively folded or unfolded (αH ↔ 3$_{10}$-helix ↔ β-strand ↔ turn ↔ bend ↔ coil), and their overall folding content, estimated on the average structure, is close to that of the KIT$^{WT}$, except A-loop (Figure S3; Table S1). The effect caused by the D816V mutation on A-loop folding is in concordance with the early reported observations in a KIT$^{D816V}$ partial construct, studied by MD simulation [26]. In particular, the A-loop helical folding increased significantly in KIT$^{D816V}$ (13%) compared to KIT$^{WT}$ (3%), mainly through a folding up- or downstream from the point mutation. At the same time, the A-loop β-hairpin in KIT$^{D816V}$ is decreased compared to KIT$^{WT}$. Consequently, according to the Define Secondary Structure of Proteins (DSSP) assignment, D816V mutation significantly influences A-loop folding only. However, an estimation of the difference in the probability of formation of secondary structures for each pair of residues *i* from KIT$^{WT}$ and KIT$^{D816V}$ showed that the effects caused by the D816V mutation on KIT folding are observed in other remote regions from the point mutation site, and these effects are more pronounced in the IDRs—JMR KID and C-tail—and even in well-structured αC-helix.

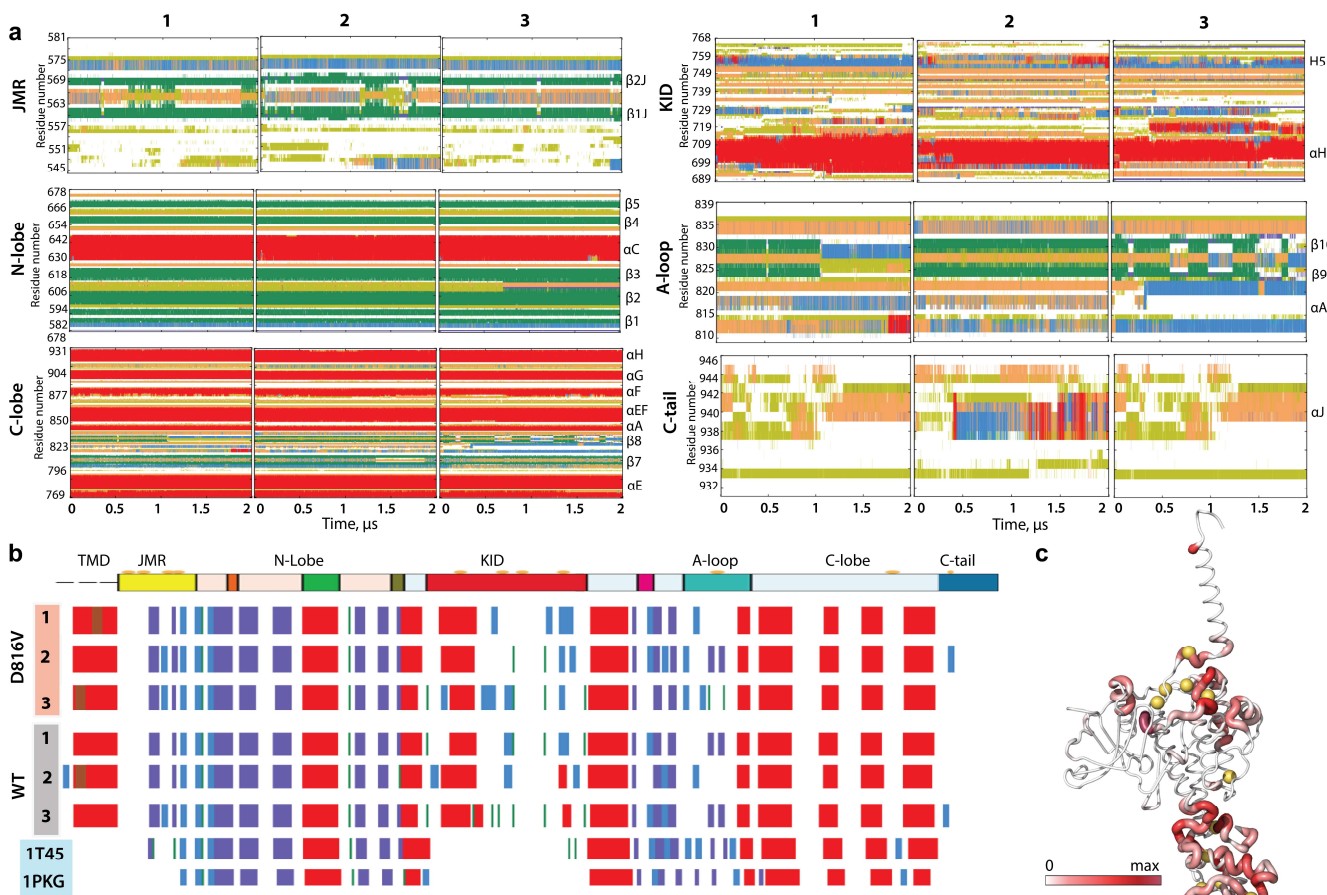

**Figure 2.** KIT$^{D816V}$ folding. (**a**) The time-dependent evolution of the secondary structures of KIT$^{D816V}$ per domain/region, as assigned by the Define Secondary Structure of Proteins (DSSP) method: α-helices in red, 3$_{10}$-helices in blue, parallel β strands in green, antiparallel β strands in dark blue, turns in orange, and bends in dark yellow. The three MD replicas (1–3) were analysed individually. (**b**) The secondary structures—αH-(red), 3$_{10}$-helices (light blue), and β-strands (dark blue)—assigned for a mean conformation of each MD trajectory (1–3) of KIT$^{D816V}$ and KIT$^{WT}$, and for the crystallographic structures of inactive (PDP ID: 1T45) and active (PDB ID: 1PKG) KIT$^{WT}$. (**c**) The increase in helical folding of KIT$^{D816V}$ compared to KIT$^{WT}$ was estimated by the difference in probability of folding per couple of residues *i* from KIT$^{WT}$ and KIT$^{D816V}$ and illustrated by the thickness of the coloured ribbons (large ones are red, small ones are white).

## 2.4. KIT Plasticity: Mutation-Induced Effects on the Conformational Space

Besides the IDRs' reversible transitions at the folding level, conformational plasticity KIT$^{D816V}$ appears from mutual linear and rotational displacements of local structures in each ID region, or/and global displacements of regions/domains, both ordered and disordered, relative to each other. These effects also lead to an intrinsic disorder at the conformational level.

To capture the conformational heterogeneity of KIT$^{D816V}$, and, particularly, to compare the conformational spaces of KIT$^{D816V}$ and KIT$^{WT}$, we analysed the conformational ensemble of each KIT by Principal Component Analysis (PCA), a multivariate statistical technique employed to reduce the dimensionality of large datasets, leading to increase in interpretability and minimising information loss [50]. This method systematically reduces the number of dimensions required to describe protein dynamics through a decomposition process that filters observed motions from the largest to the smallest spatial scales [51].

Before conducting a comparative analysis between KIT$^{D816V}$ and KIT$^{WT}$, it is worth noting that the three first modes describe about 85–90% of all collective motions observed in each of the three KIT$^{D816V}$ MD trajectories (Figure 3a).

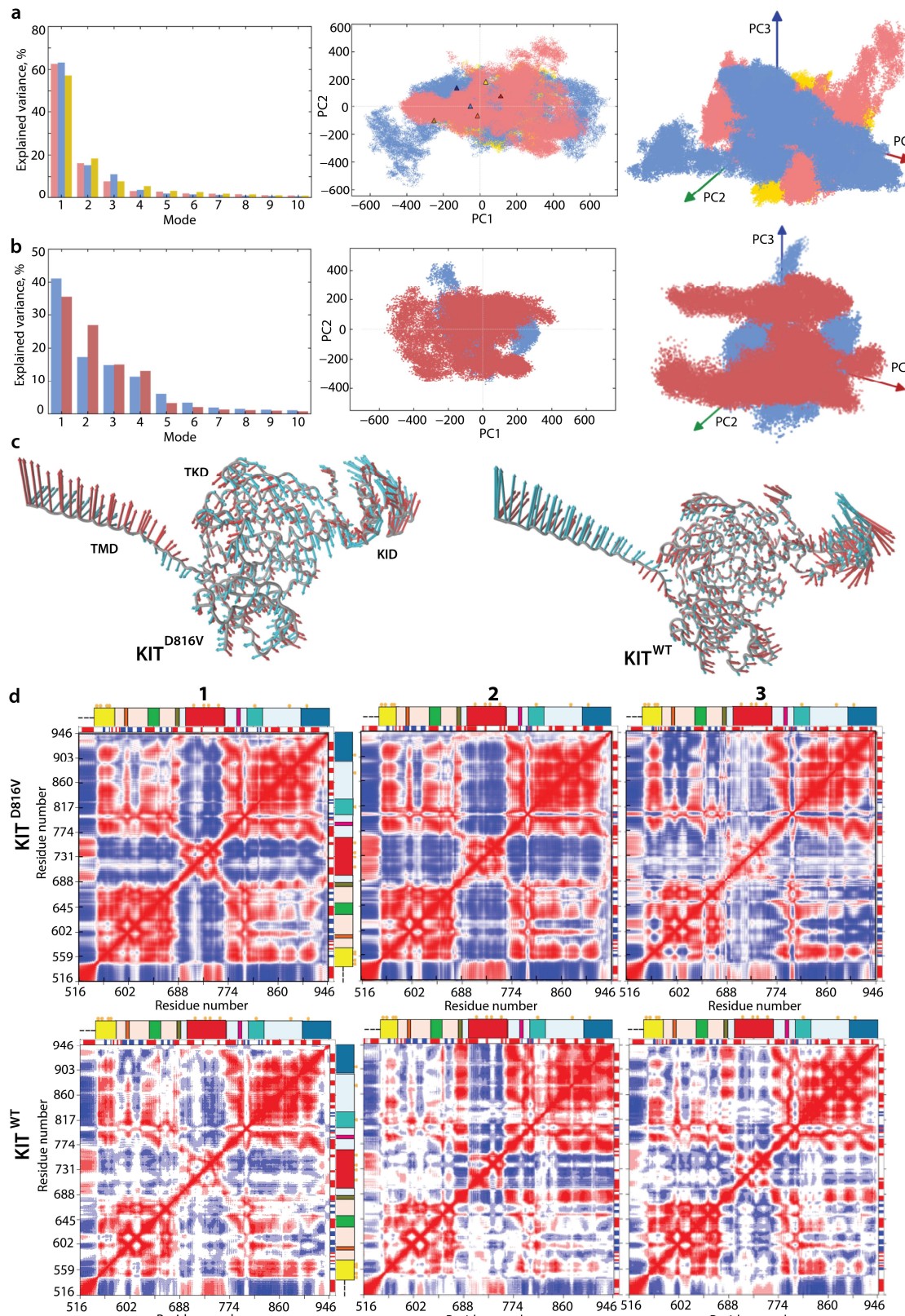

**Figure 3.** Intrinsic motion of KIT and its interdependence. (**a**) PCA modes calculated for each trajectory of KIT$^{D816V}$ after least-square fitting of the MD conformations to the initial conformation (t = 0 μs). The bar plot gives the eigenvalue spectra, in descending order, for the first 10 modes calculated

on MD trajectories 1–3 (**left**). Projection of the KIT$^{D816V}$ MD conformations onto the first two (**middle**), and three PCA modes (**right**). MD trajectories 1–3 are denoted in red, blue, and yellow, respectively. Light and dark symbols display the first and the last conformations for each trajectory. (**b**) PCA modes calculated for concatenated trajectories of KIT$^{D816V}$ (red) and KIT$^{WT}$ (blue) after least-square fitting of the MD conformations to the initial conformation (t = 0 μs). The bar plot gives the eigenvalue spectra, in descending order, for the first 10 modes (**left**). Projection of the KIT$^{D816V}$ (red) and KIT$^{WT}$ (blue) MD conformations onto the first two (**middle**) and three PCA modes (**right**). (**c**) Atomic components in PCA modes 1 and 2 are drawn as red (1st mode) and cyan (2nd mode) arrows, projected on the cartoon of KIT$^{D816V}$ (**left**) and KIT$^{WT}$ (**right**). A cut-off distance of 4 Å was used. (**d**) Dynamical inter-residue cross-correlation map computed for the Cα-atom pairs of MD conformations of KIT$^{D816V}$ (**top**) and KIT$^{WT}$ (**bottom**) from each individual trajectory (1–3). Correlated (positive) and anti-correlated (negative) motions between Cα-atom pairs are shown as a red–blue gradient.

The projection of KIT$^{D816V}$ MD conformations from three independent replicates onto the two or three first principal components (PCs) shows essential overlap. The overlapping areas are formed by the conformations reproduced in the replicates and consist of statistically rich 'cloned' data, while the non-overlapping areas represent novel conformations observed in only a single trajectory that complement the 'cloned' data, yielding a generic dataset. Obviously, the generic conformational space combined from the generated subspaces does not represent a complete conformational space of the disordered KIT but more exhaustively reflects its conformational properties than a single subspace.

The extensively overlapped three KIT$^{D816V}$ MD replicates, as justified by PCA, supports their merging and further calculations on the concatenated trajectories. Moreover, since KIT$^{WT}$ MD trajectories were also highly comparable [43], we used these conformational generic ensembles characterising two different proteins, KIT$^{D816V}$ and KIT$^{WT}$, to make a comparison between them.

Surprisingly, the conformational sets of two proteins show a large overlap, suggesting, in the first approximation, a good similarity of their conformational spaces (Figure 3b). The first two PCA principal components, projected onto the KIT 3D structure, clearly reflect highly coupled motions of the each multidomain KIT (Figure 3c). The significant mobility of the TM helix is gradually increased from its C- to N-ends in both proteins, a phenomenon often observed in membrane proteins [52–54]. The global motion in the TK domain of both KITs, showing less amplitude than the TM helix and KID, is highly collective and described as pendulum-like circular movement along a common virtual axis of rotation. Interestingly, this virtual rotational axis coincides with the active site of each KIT. Surprisingly, the TKD collective motions calculated on the concatenated trajectories of KIT$^{D816V}$ and KIT$^{WT}$ are comparable in amplitude and differ only in direction, revealing the better circularity of KIT$^{WT}$ compared to KIT$^{D816V}$. Analysis of individual trajectories better demonstrates this directional difference in the TKD motion of two proteins (Figure S4).

The intrinsic dynamics of the two multidomain KIT were analysed with the cross-correlation matrix. The matrices computed for each Cα-atom pair are very similar for the three MD trajectories of each protein, but they differ greatly between the proteins (Figure 3d). In both proteins, the cross-correlation map demonstrates highly coupled motions within each KIT domain and between the domains, even those that are largely distant in sequence and 3D structure. Nevertheless, the cross-correlation pattern in KIT$^{D816V}$ is systematically much more contrasted in respect to KIT$^{WT}$, and, consequently, reflects the more correlated motion.

The matrix pattern in the N-lobe of each protein reflects the positively correlated movement of the seven stands in the crossed β-pleated sheet and their coupling with the αC-helix. Similarly, the C-lobe helices are mutually correlated (positively), forming a pattern of well-defined blocks distorted by the A-loop. The two TK lobes are positively correlated with each other, negatively with the KID and TM helix, and positively with JMR, and these correlations are significantly stronger in KIT$^{D816V}$.

The intra- and inter-lobe correlations are also not identical in KIT$^{D816V}$ and KIT$^{WT}$. The N-lobe of KIT$^{D816V}$ is represented by a more homogeneous block, reflecting highly collective motions of P-loop and αC-helix with JMR. Since coupled motion reflects allosteric regulation of functionally relevant fragments involved in KIT activation mechanisms and post-activation processes, the observed differences in inherent motions and mutual correlations of two proteins strongly indicate a more active KIT$^{D816V}$ compared to KIT$^{WT}$.

The motions of TM helix and JMR are contrariwise in both proteins. Such correlation patterns can be partially explained by the overall architectural features of the studied KIT proteins, which have a strongly extended shape. Movements of one end (TM helix) are counterbalanced by movements of the opposite end (KID) to provide a stable balance of KIT around its centre of gravity.

### 2.5. Impact of D816V Mutation on Inter-Domain Non-Covalent Interactions Stabilising KIT

To examine the impact of D816V mutation on the non-covalent interactions stabilising KIT 3D structure, we first compared H-bond patterns in KIT$^{D816V}$ and KIT$^{WT}$. A large number of stable (long-lived) H-bonds were observed in both proteins (Figure 4a).

The main difference consists of the H-bond pattern formed by the A-loop and its ends: A-loop in KIT$^{WT}$ is linked by strong and multiple H-bonds with the C-loop, JMR, and the C-lobe residues close to the C-tail, whereas, in KIT$^{D816V}$, the A-loop is strongly bound by its N-term only with the linker connecting C-helix and Hinge; moreover, its contacts with C-loop are weakened. The JMR in KIT$^{D816V}$ is stabilised only by H-bonds with C-loop and C-lobe residues adjacent to the A-loop C-end.

Focusing on the active site and closed-to-active site residues, we observe that the H-bond pattern in KIT$^{D816V}$ is significantly changed compared to KIT$^{WT}$ (Figure 4b). Thus, A-loop D820 forms a salt bridge with C-loop R796, and residue Y823 links by H-bond R815, stabilising two A-loop sub-fragments and attaching D792 from C-loop, making a strong bifurcate bonding connecting three fragments together. Further, K623 of A-loop interacts with E640 from αC-helix, forming a salt bridge. Interestingly, the H-bond contact of the JMR W557 interacts by a unique donor group N-H with the π-system of H790 from C-loop. We note that the residues D810, F811, and G812, which form the highly conserved *catalytic* (DFG) motif, are not involved in H-bonding in both proteins.

This KIT$^{D816V}$ H-bond pattern, highly differing from KIT$^{WT}$, apparently plays a crucial role in the constitutive activation of the oncogenic mutant. Surprisingly, the H-bond between the A-loop residue Y823 and C-loop E792, interpreted as a key interaction maintaining the allosteric pathway between A-loop and JMR in the inactive native protein [25], is observed in the constitutively active KIT$^{D816V}$.

Despite the close proximity of A-loop and JMR in the 3D structure of the inactive KIT$^{WT}$ characterising the autoinhibitory position of JMR, there are no direct H-bonds between these two fragments, as shown by the analysis of the crystallographic structure (PD ID: 1T45) and short MD simulations. A-loop and JMR interact only indirectly through the C-lobe residues as connecting intermediates. Similarly, in the constitutively active KIT$^{D816V}$, the A-loop and JMR also interact thought intermediates linkers, which are different from KIT$^{WT}$.

Generally, the TKD, a core structure of KIT, shows remarkable stability in a given state of the protein, the wild type inactive (KIT$^{WT}$) or constitutively active (KIT$^{D816V}$), and these states are stabilised by state-specific H-bond patterns. During the transition from the inactive to the active state, numerous intermediate metastable conformations of KIT are apparently observed. Indeed, the detected collective motions in each protein favour a large number of very different KIT conformations. Such conformational heterogeneity mainly involves functional KIT regions possessing the tyrosine residues—JMR, A-loop, KID, and C-tail. Similar to KIT$^{WT}$, the phosphorylation site positions in KIT$^{D816V}$ strongly depend on their host fragments, differing by conformational properties and relative position in the protein.

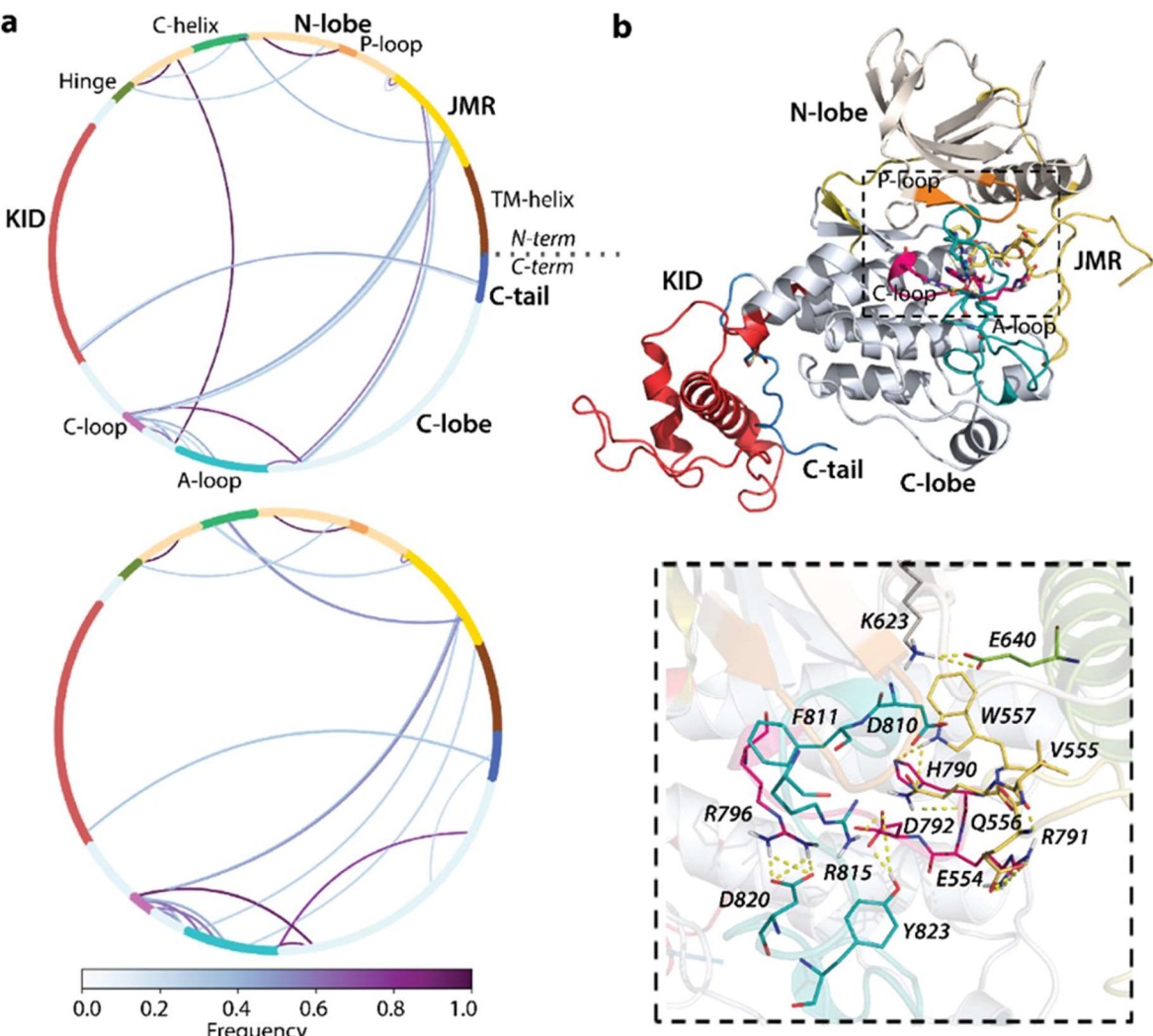

**Figure 4.** Hydrogen bond pattern stabilising RTK KIT. (**a**) The cord diagram compiles the H-bonds of multidomain KIT[D816V] (**top**) and KIT[WT] (**bottom**). H-bonds are shown as curves, coloured according to occurrences, from 0 (white) to 100% (purple). (**b**) H-bonds stabilising the KIT[D816V] active site (**top**) are shown by yellow dashed lines (**bottom**). Protein is shown as cartoon, with domains and functionally related fragments distinguished by colour and labelled in bold and regular font, respectively. The active site and neighbouring residues contributing to H-bonds are shown as sticks and labelled in italics. Calculations are performed on the concatenated trajectories.

Thus, the extended displacement (linear and rotational) of the structural fragments within the KID, and the KID as a 'pseudo-rigid body' relative to the TK domain, is reflected in the expanded position distributions of the Cα-atoms and hydroxyl groups of Y721, Y747, and Y730, located on the highly flexible fragments of the disordered KID, while the OH groups of Y703 in the stable αH1-helix form a narrower cluster, mainly resulting from the global rotational displacement of the KID relative to TKD (Figure 5).

Likewise, the compact distribution of the Y553, Y568, and Y570 locations, viewed by the Cα-atoms and the OH groups, reflects the stable position of JMR fragments—JMR-Binder, JM-Zipper, and JMR-Switch—while the wide-ranging distribution of Y547 location corresponds either to the multiple inherent JMR conformations or ample displacement of JM-Proximal, with respect to the TK domain. In KIT[D816V] the compact single cluster, representing a unique phosphotyrosine Y823 of A-loop, differs by subdividing into three different clusters in KIT[WT]. The JMR Y547 shows enlarged distribution in the two orthogonal directions in respect to the one-direction displacement in KIT[WT]. This Y547 distribution

is apparently derived from the spatial linear and rotational displacement respective to TKD, similarly to the KID tyrosine residues Y721, Y747, and Y730.

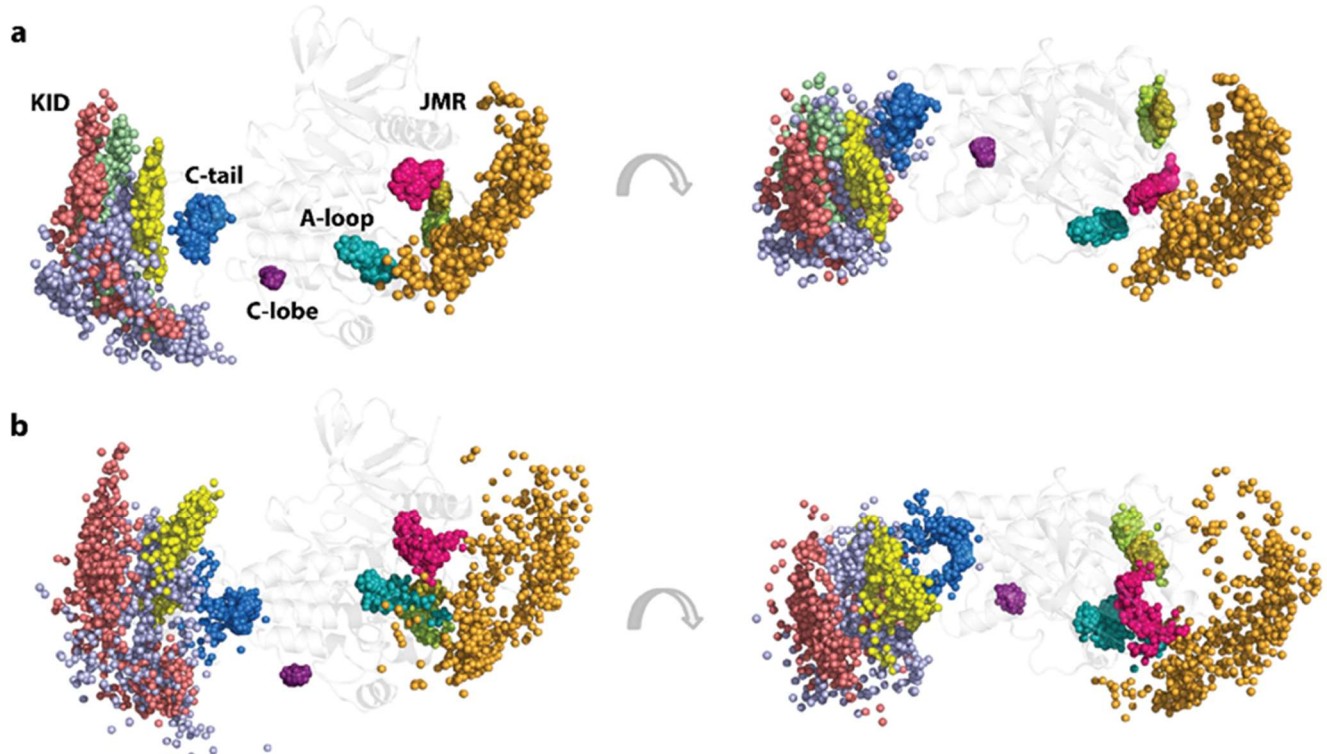

**Figure 5.** Geometry of the tyrosine residues in KIT$^{D816V}$. The spatial distributions of the tyrosine residue's Cα-atoms (**a**) and hydroxyl groups (OH) presented by the oxygen (O) atoms (**b**), are shown in two orthogonal projections. The Cα- and O-atoms of different tyrosine residues are distinguished by colour: Y547 in orange, Y553 in magenta, Y568 in smith green, Y570 in lime, Y703 in yellow, Y721 in lilac, Y730 in red, Y747 in green, Y823 in teal, Y900 in purple, and Y936 in blue. The Cα- and O-atom positions were extracted from the MD conformations, taken each 10 ns, fitted on the TKD of the initial structure (t = 0 ns), and superimposed on this structure countered in grey.

Each of the other tyrosine residues, Y900 and Y936, located, respectively, in C-lobe and C-tail, forms a unique cluster, either very compact (Y900) or slightly enlarged (Y936). Their positions and compactness are similar to those observed in KIT$^{WT}$.

## 2.6. Per Domain Clustering of KIT$^{D816V}$ Conformations

To characterise the inherent structural and conformational properties of the highly variable KIT$^{D816V}$ regions, i.e., JMR, KID, A-loop, and C-tail, and estimate the contribution of these properties to the global dynamics of protein, each of these fragments was analysed individually. First, the conformations of each fragment were grouped by ensemble-based clustering [55] using various RMSD cut-off values, varying from 2.0 to 4.0 Å with an increment of 0.5 Å. A cut-off value of 4 Å was used for JMR, KID, and C-tail, while a cut-off value of 2 Å for A-loop was sufficient to regroup the conformations of all fragments into clusters that give the best cumulative population (>95%).

The majority of JMR conformations are encompassed within the three most populated clusters, C1 (52%), C2 (19%), and C3 (12%), comprising conformations observed in each MD trajectory (Figure 6). All JMR conformations exhibit transient secondary structures, a short β-sheet, and a transient 3$_{10}$-helix in the JM-S and JM-Z segments, which undergo reversible folding/unfolding events. The representative conformations of these clusters differ only in the position of the JM-P segment containing Y547, previously identified in vitro as a

phosphotyrosine [56]. This alternate conformational dispersion of JM-P leads to a large area of Y547 location, while the other tyrosines are nearly superimposed (Figure 5).

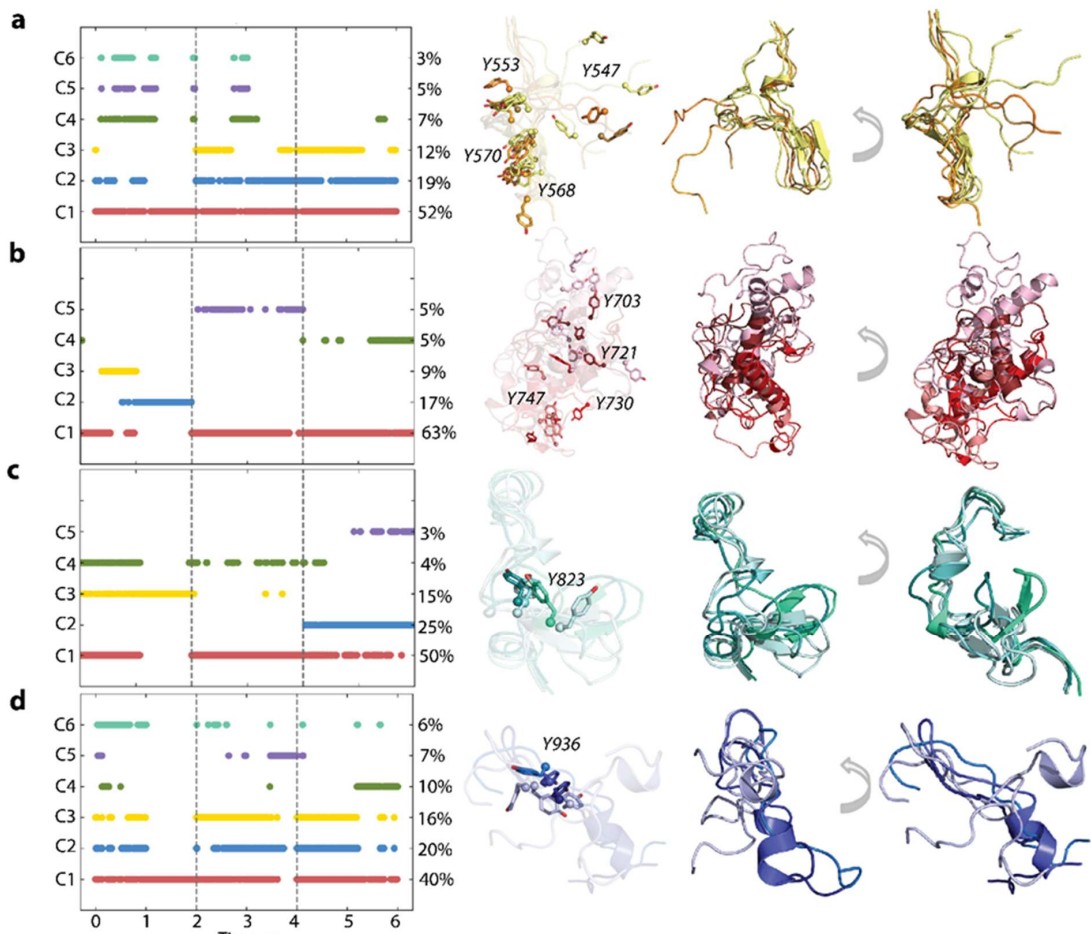

**Figure 6.** Structure and conformation of the disordered fragments of KIT$^{D816V}$—JMR, KID, A-loop, and C-tail. (**a–d**) The clusters obtained by ensemble-based clustering (cut-off of 4 Å for JMR, KID, and C-tail, and 2 Å for A-loop) and their population (**left column**). Superimposed representative conformations from the clusters (**right column**). The protein is shown as cartoon, tyrosine residues as sticks. The colour gradient shows the population of clusters, from dark (most populated) to light (less populated). Tyrosine residue numbering is shown for the most populated cluster. Calculations were carried out on the MD conformations of concatenated trajectory, taken every 100 ps after fitting on the initial conformation of the TKD (at t = 0 µs).

The MD conformations of the intrinsically disordered KID were grouped into five clusters. Only cluster C1 (63%) comprises conformations generated across all trajectories, while the remaining clusters consist of conformations from individual trajectories. The representative conformations from these distinct clusters exhibit variation in folding (2D) and 3D structure organisation, highlighting the substantial intrinsic disorder of KID. The intra-KID extensive rotationality and linearity of its structural units contribute to the large conformational diversity of KID, resulting in dispersed locations of the tyrosine residues, illustrated in Figure 5.

Focusing on A-loop, we noted that the most populated cluster (C1, 50%), composed of conformations from all trajectories, contains the well conserved β-hairpin and small $3_{10}$-helix transient helix (Figure 6). Conformations from the other, less populated, clusters show decreased folding ($3_{10}$-helix ↔ turn) and displaced β-hairpin. The more dispersed location of Y823 in KIT$^{D816V}$ is derived from different A-loop conformations (Figure 5). The conformations of C-tail are grouped into the three most populated clusters, C1 (40%),

C2 (20%), and C3 (16%), and three sparsely populated clusters, C4-C6 (10–6%) (Figure 6). The representative conformations display similar secondary structures, described as an extended random coil with a small transient helix in the C-tail middle ($\alpha$-helix $\leftrightarrow$ $3_{10}$-helix $\leftrightarrow$ coil). The differences lie mainly in the orientation of the C-end residues. All clusters regroup conformations generated across the three independent trajectories, and, apparently, C-tail secondary structures do not influence a cluster separation. The tyrosine residue Y936 shows a dense cluster and orientation of its OH group in clusters 1–3, which differs in only a few rare intermediate conformations (Figure 5).

This cluster analysis of the KIT$^{D816V}$ individual ID regions demonstrates that the degree of the intrinsic structural and conformational disorder is greater compared to KIT$^{WT}$ [43], highlighting the mutation-induced effects on each functional KIT fragment—JMR, A-loop, KID, and C-tail. The comparative analyses of ID regions in KIT$^{D816V}$ and KIT$^{WT}$ indicate that KID is the most disordered domain and the most influenced by the D816V mutation.

We note that such an analysis only reflects the inherent characteristics (intrinsic disorder) of the KIT IDRs, which were considered in the calculation as 'cleaved' species, while their contribution to the extrinsic disorder describing global plasticity (dynamic inter-domain/region relationships) in both proteins is not yet revealed.

### 2.7. What Are We Learning from the Cumulative Free Energy Landscape of KIT$^{D816V}$ and KIT$^{WT}$?

In the above comparative analysis of their structural and conformational properties, the multidomain KIT$^{WT}$ and KIT$^{D816V}$ consist of a relatively stable TK domain surrounded by four intrinsically disordered regions—JMR, KID, A-loop, and C-tail.

To describe the conformational spaces of this type of disordered protein exhibiting reversible local folding–unfolding and local/global conformational diversity, and to compare between them, we employed an explicit 'free energy landscape' model. Such interpretation of intrinsically disordered proteins yields quantitatively significant results, enabling comparisons between the multidomain protein in different states. The relative Gibbs free energy, $\Delta G$, defined on chosen coordinates, called 'reaction coordinates' or 'collective variables', describes the conformations of a protein between two or more states, quantified by the probability of finding a system in those states. Such representations of the disordered protein sampling, utilising reaction coordinates, can serve as a quintessential model system for barrier-crossing events in such proteins [57].

We first evaluated the $\Delta G$ of KIT$^{D816V}$ and reconstructed its landscape using distant measures—radius of gyration ($Rg$), distance (RMSD), and the PCA components (PC1 and PC2)—as reaction coordinates. The free energy landscape ($FEL$) as a function of RMSD and radius of gyration $Rg$ ($FEL_{RMSD}^{Rg}$) and of the PCA components ($FEL_{PC1}^{PC2}$) calculated for the concatenated replicas of KIT$^{D816V}$ is shown in Figure 7.

Similar to KIT$^{WT}$, the KIT$^{D816V}$ FELs show a rugged landscape, typical for intrinsically disordered proteins. Nevertheless, both *FELs* of KIT$^{D816V}$ display areas of minimum energy that represent more probable conformations (the thermodynamically more favourable states), while the reddened areas indicate transient conformations of the protein (state-to-state transition). The free energy landscape generated as a function of RMSD and radius of gyration $Rg$ ($FEL_{RMSD}^{Rg}$) exhibits two deep minima (wells), W1 and W2, separated from each other by an energy barrier. At very close proximity to W1, W3 was localised. Looking the well's content, we found that W2 and W3 are composed of very similar conformations in JMR, KID, and C-tail positions respective to TKD, the H1 position in KID, and differed mainly in A-loop folding. In contrast, conformations of KIT$^{D816V}$ from W1 exhibit positions of JMR, KID, and C-tail very dissimilar to those from W1 and W2. The probability of the conformations observed in W1 is almost comparable to the combined probability of conformations from W2 and W3.

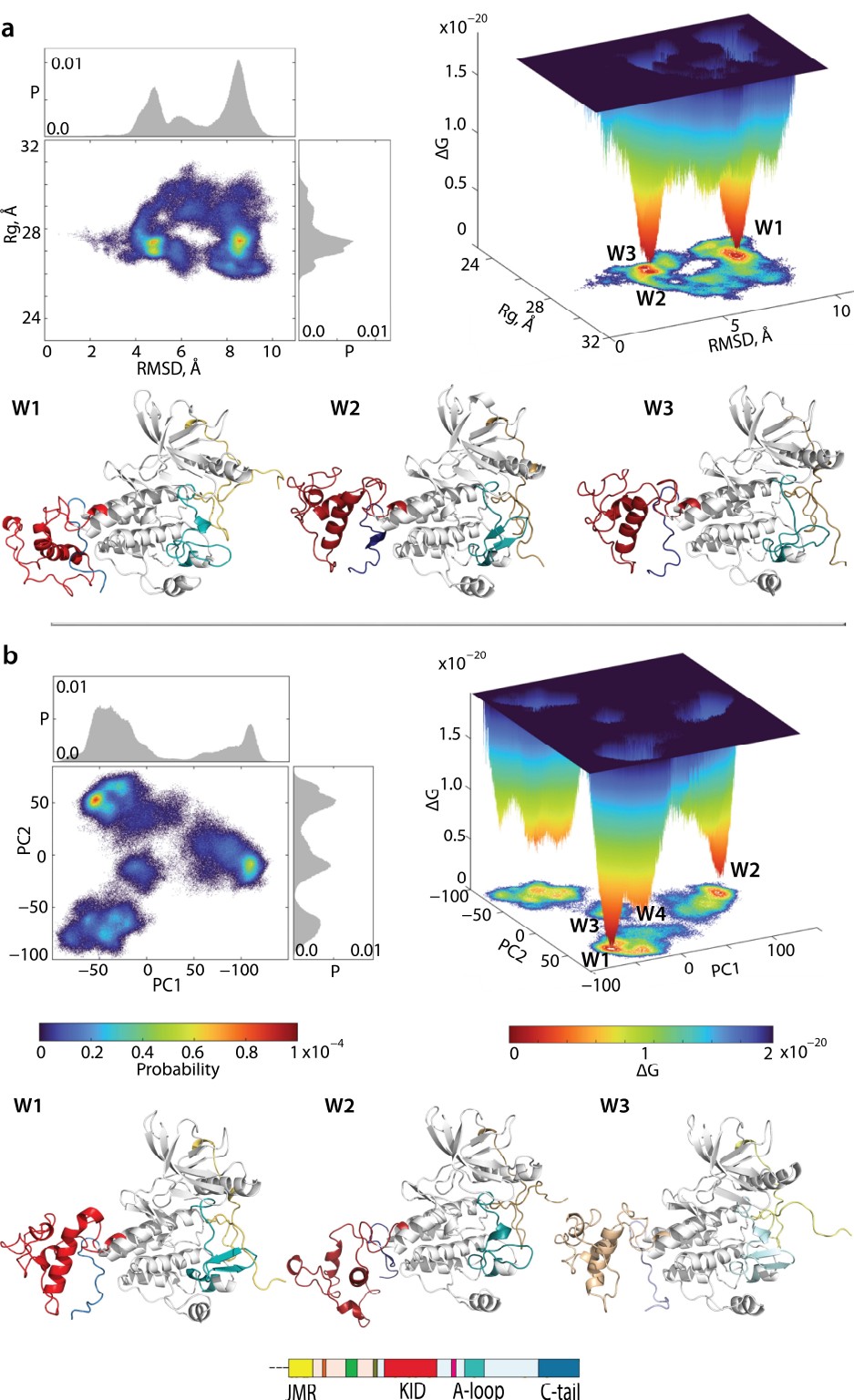

**Figure 7.** Free energy landscape (*FEL*) of KIT[D816V] as a function of the reaction coordinates, (**a**) $R_g$ (in Å) versus RMSD (in Å) and (**b**) two PC components (PC1 versus PC2) were generated using the MD conformations of KIT[D816V] for the conformational ensemble sampled from the merged replicas and fitted on the initial conformation. (**top panel**) The 2-dimensional representation of the FEL of the KIT[D816V] conformational ensembles (**left**). Probability distribution of each reaction coordinate is shown at the top and right, respectively. The 3-dimensional representation of the relative Gibbs free energy (**right**). The red colour represents high occurrence, yellow and green low, and blue represents the lowest occurrence. (**bottom panel** (**a**,**b**)) KIT conformations from wells 1–3.

As the PCA principal components PC1 and PC2 explain a large portion of variance (48 and 31%, respectively), these metrics were used as the second pair of reaction coordinates. The FEL designed on these metrics ($FEL_{PC1}^{PC2}$) is a complex landscape displaying multiple wells. Instead of two well-separated minima in KIT$^{WT}$, in KIT$^{D816V}$ we observe a series of minima of different depths that are either significantly distant (W1, W2, and W4) or very close (W1 and W3). The most populated W1 and its satellite W3 are composed of similar conformations deferring only in the length of the KID H1-helix, which is elongated in W1 and shortened in W3. In conformations from W2, the KID H1-helix is oriented perpendicularly to the helices from C-lobe. Such orientation probably favours C-tail insertion between KID and C-lobe. Conformations from W4 show KID folding similar to that in W3 but positioned closely to C-lobe of TKD.

Since the two proteins, KIT$^{D816V}$ and KIT$^{WT}$, differ only in the point mutation (D816V), which can be considered as an internal effector promoting irreversible structural and conformational changes in the mutant, we hypothesise that the cumulative representation of the two conformational spaces as a 'free energy landscape model' may shed light on key differences/similarities between the two proteins, acknowledging certain limitations to such a comparison.

For the evaluation of the relative free energy ($\Delta G$) and the reconstruction of its landscape, we employed the PCA components (PC1 and PC2) as reaction coordinates. The free energy cumulative landscape (FECL) as a function the PCA principal components was computed for the concatenated trajectories combining MD conformations of both KITs normalised on a common reference structure (KIT$^{WT}$, at t = 0 μs).

Each 2D FECL exhibits a rugged scatter plot, reminiscent of the artistic technique known as 'pointillisme', which employs small, juxtaposed areas of colour, reflecting a high conformational heterogeneity of the analysed conformations (Figure 8).

The observed cloud-like scatter plot of the multidomain proteins KIT$^{WT}$ and KIT$^{D816V}$, built on the PCA two first principal components, arises due to two major factors: (i) the large inherent conformational divergence in each protein, each possessing at least four intrinsically and extrinsically disordered regions, and (ii) the proximity of the structural and conformational properties of the TK domain in the two proteins, promoting partial overlap of their conformational spaces. This complexity presents challenges in interpreting the free energy map.

Nevertheless, the 3D FECL of two KITs show either smooth or well-defined local areas of minimum energy, indicated in red, which represent more stable conformations (the thermodynamically more favourable state). The reddened areas indicate conformational transitions of the proteins.

The FECL of the full-length cytoplasmic domain of both KIT has multiple minima, wells labelled W1-W5, separated by energy barriers of various heights. W1 and W5 consist of a bi-component non-equivalent content, comprising MD conformations of KIT$^{WT}$ (73 and 78%, respectively) and KIT$^{D816V}$ (27 and 22%, respectively). W2 is exclusively composed of KIT$^{WT}$ conformations (100%), while W3 and W4 predominantly include KIT$^{D816V}$ conformations (87 and 96%, respectively). The W1 and W2 KIT$^{WT}$ conformations significantly differ in several aspects, including (i) KID orientation in respect to TK domain, (ii) A- and C-loop folding, (iii) C-tail folding, and (iv) JMR conformations. The major component of W5 is KIT$^{WT}$ conformations, in particular KIT fragments, either resembling their W1 (C-loop) or W2 (A-loop and C-tail) counterparts or differing from them (KID). The KIT$^{D816V}$ conformations, dominant in W3 and W4, differ mainly from KIT$^{WT}$ conformations in KID and JMR folding, as well as in JMR and C-tail conformations. Other structural/functional fragments are similar to KIT$^{WT}$, either in W1 (A-loop) or W2 (C-loop) or W5 (JMR conformation).

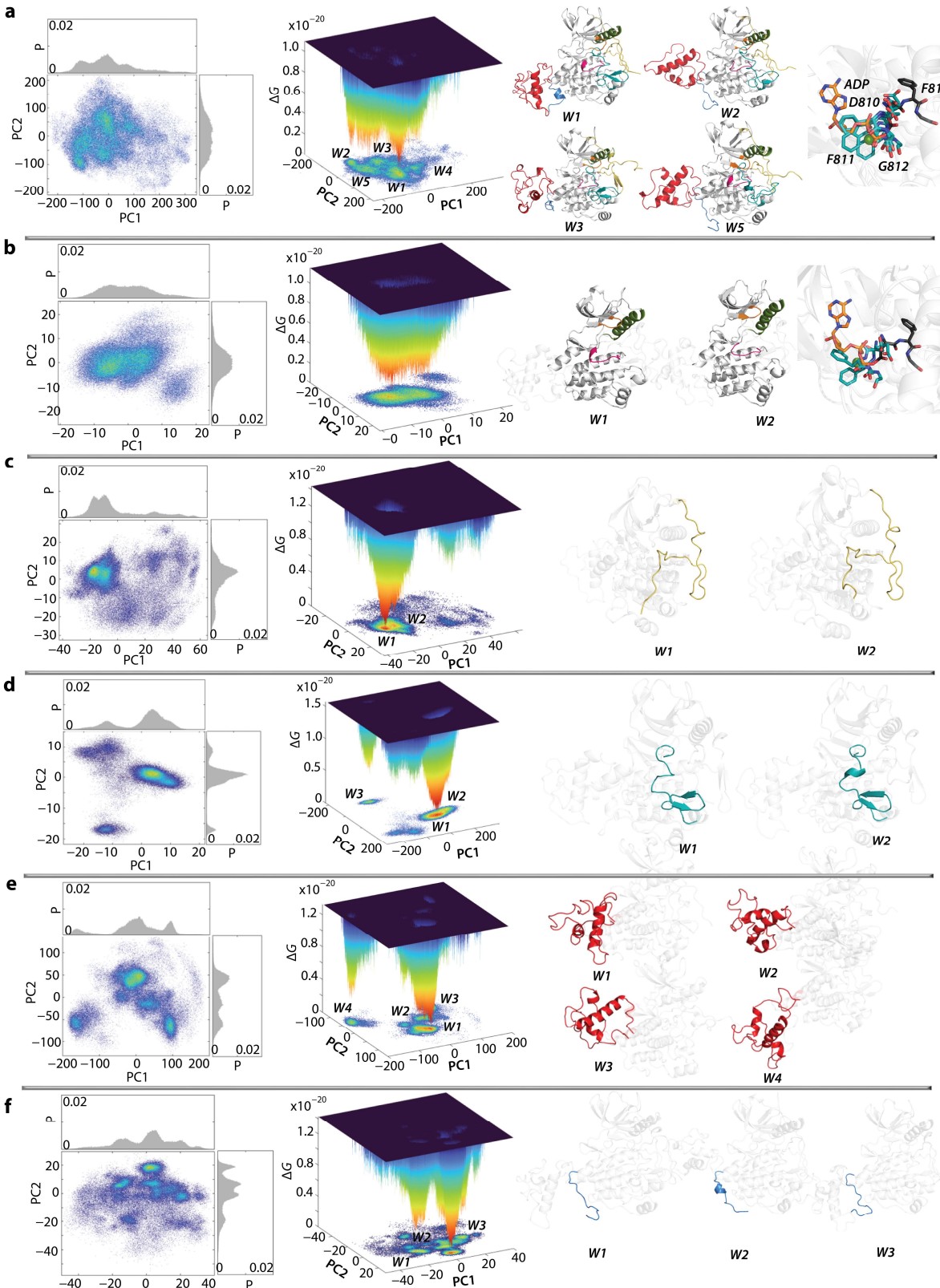

**Figure 8.** Free energy cumulative landscape (FECL) of two KIT proteins as a function of the reaction coordinates, taken as two PCA components (PC1 vs PC2). FECLs were generated on the concatenated trajectories consisting of MD conformations of KIT$^{WT}$ and KIT$^{D816V}$ mutant, normalised to the initial conformation of KIT$^{WT}$ (t = 0 μs). (**a–f**) The 2D representation of FECL of the KIT conformational ensembles

focusing the full-length cytoplasmic domain (**a**); the full-length cytoplasmic domain without A-loop (**b**); the JMR (**c**); the A-loop (**d**); the KID (**e**); and C-tail (**f**) (**left column**). The 3D representation of the relative Gibbs free energy (**middle column**). The red colour represents the highest occurrences, yellow and green represent low, and blue represents the lowest occurrences. The wells (W) are numbered from deepest (W1) to shallower (W2, W3,...). Each identified well contains at least 1% of the total conformations, and $\Delta G \leq 0.45 \times 10^{-20}$ $K_b T$. The major representation conformation (in cartoon) from the most populated minima (**right column**) is highlighted by colour: white (TKD or not relevant domain), yellow (JMR), orange (P-loop), green (αC helix), red (KID), pink (C-loop), A-loop (teal), C-tail (blue).

In all MD conformations of KIT$^{WT}$ and KIT$^{D816V}$ within the lowest energy well, the conserved *catalytic* (DFG) motif is in the DFG-in conformation, while, in the KIT$^{WT}$ active state, the DFG-out configuration is observed (PDP ID: 1PKG).

Construction of FECL of the full-length cytoplasmic domain without A-loop yields to two local minima, W1 and W2, with a two-component content. W1 is composed of KIT$^{WT}$ (60%) and KIT$^{D816V}$ (34%) making a 2:1 population ratio, while W2 is primarily composed of KIT$^{D816V}$ (90%). Representative major conformations of W1 and W2 differ in the folding of C- and P-loop and the relative position of the P-loop and αC-helix. An observable approach of P-loop toward αC-helix is seen in KIT$^{D816V}$, but not to the extent observed in the crystallographic structure of the active KIT$^{WT}$ (PDB ID: 1PKG).

The D816V mutation appears mostly to affect αC-helix orientation and increase C-loop fold in its N-end containing the catalytic residue D792 from the conserved *catalytic* (DFG) motif, which is in DFG-in conformation in both major conformations.

As the PCA of the full-length CD reflects all structural and dynamic effects together, (i) reversible folding–unfolding ID regions, (ii) local (intrinsic) flexibility, and (iii) positions of IDRs relative to the kinase domain derived from extrinsic disorder (global plasticity), it remains challenging to definitively determine the D816V effects on each individual domain of KIT.

To achieve a thorough understanding of short-range and distant mutation-induced effects on each IDR, a deeper analysis of each sub-domain analysis is necessary. As such, to eliminate the influence of other parts of the protein, PCA was carried out on each sub-domain separated from the structure and normalised by a fitting on the TK domain.

Plotting the JMR FECL revealed two adjacent minima, W1 and W2, both mainly composed of KIT$^{WT}$ (84% in W1 and W2) exhibiting similar folding and conformational properties. KIT$^{D816V}$ constitutes only a minor component (16%). In both KIT$^{WT}$ representative conformations, JMR is unfolded and is maintained in an auto-inhibition position, differing only in the position of JM-P and JM-B relative to the TK domain. Comparing KIT$^{WT}$ and KIT$^{D816V}$ conformations, it seems that JMR is strongly affected by the mutation. As was evidenced by clustering, JMR displays a greater conformational variability in KIT$^{D816V}$ (Figure 6). This large conformational dispersion of JMR leads to its highly diffused distributions in the FECL (Figure 7). In both proteins, the JM-B segment of JMR is either partially folded as a β-hairpin or fully unfolded (coiled hairpin). Regarding its position, JMR is either adheres to the TK domain (KIT$^{WT}$) or is exposed to the solvent (KIT$^{D816V}$).

The KID FECL is characterised by a series of distant minima (W1-W4). These wells are either bi-component (W1 and W2), with almost equal composition (W1 contains 53% of KIT$^{WT}$ and 47% of KIT$^{D816V}$), or predominantly one (W2 contains 81% of KIT$^{D816V}$ and 19% of KIT$^{WT}$), or entirely one-component (W3 and W4 contain 100% KIT$^{WT}$ and KIT$^{D816V}$, respectively).

Comparing KID conformations of KIT$^{WT}$ forming wells W1 and W3, their main difference pertains to the orientation of the αH1-helix and the overall KID conformation. Similarly, KID of KIT$^{D816V}$ from W2 and W4 differ in the orientation of the helices.

However, in all wells, KID retains a compact globular shape in both KIT$^{WT}$ or KIT$^{D816V}$, with all αH1, $3_{10}$-H3, and αH5-helices oriented differently relative to each other. When comparing KID of the two proteins, the same difference in folding level becomes evident,

and D816V mutation seems to have affected the overall position of KID relative to the TK domain and the length of helices, but not their number.

As the carrier of the mutation at position 816, A-loop remains the most attractive fragment on which to evaluate the short-range effect of the mutation. The FECL plot of A-loop shows two widely separated minima, W1 and W2, both with a bi-component nature. W1 and W2 contain the KIT$^{WT}$ (76%) and KIT$^{D816V}$ (24%) conformations, with W2 mainly comprising KIT$^{D816}$ (85%) and only 15% of KIT$^{WT}$.

The difference between the A-loop from the two wells relates more to its folding than orientation with respect to the TK domain. In both wells, W1 and W2, A-loop is mostly unfolded except for β-hairpin with a variable length of its folded segment.

In addition, A-loop from W2, primarily populated by KIT$^{D816}$ conformations, shows a small $3^{10}$-helix positioned just before the mutation site. Such folding was detected on limited (50 ns) MD simulations of both proteins and interpreted as a partially transient event rather specific to KIT$^{D816V}$ rather than KIT$^{WT}$ [26].

The FECL of the highly flexible C-tail is described by a series of differently populated clusters, with three being the most crowded. All clusters are bi-component but with varying protein population. W1 is composed of KIT$^{WT}$ (91%), and W2 has nearly equal population of both proteins (52% of KIT$^{WT}$ and 48% of KIT$^{D816V}$), while W3 is dominated by KIT$^{D816V}$ (75%) and KIT$^{WT}$ (25%). C-tail of KIT$^{WT}$ shows different structures in W1 and W2, while C-tail of KIT$^{D816V}$ from W3 resembles KIT$^{WT}$ from W1, where it is unfolded and differs only by its position relative to TK domain and KID. Either it is oriented on one side of TK domain (W1) or on the opposite (W3), close to KID. In W2, C-tail show a small transient helix and its C-terminal oriented as in conformations from W1.

It is worth noting that such observations were regarding on the population of the wells, which is statistically very low compared to the large set of conformations sampled during MD simulations. Therefore, it remains challenging to identify clear structural differences between KIT$^{WT}$ and KIT$^{D816V}$, even with FECL representation on the two first principal components of a PCA. It is possible that the chosen reaction coordinates are not sufficiently suitable for the study of proteins exhibiting multiple levels of disorder (reversible folding–unfolding events, intrinsic and extrinsic plasticity) complicated by a certain number of disordered fragments. Alternatively, the length and number of replicas used may be insufficient to comprehensively capture the conformational spaces of the two KITs proteins.

## 2.8. The KIT Cytoplasmic Domain Pockets Detected in the Native and Mutated Proteins

Research into RTK inhibitors was initiated the early 1980s, when natural substances such as lavendustin A, genistein, erbstatin, and quercetin were discovered to inhibit the activities of PTKs like EGFR. This research was further continued using partial structural data in which the JMR, KID, and C-tail regions are undefined or only partially defined [49]. The catalytic ATP-binding site of the KIT, like other RTKs, has traditionally been considered the primary target of small molecules. A series of ATP competitive inhibitors targeting an inactive or active state have been approved for clinical use. The inactive state of KIT is targeted by type 2 antagonist inhibitors such as imatinib, sunitinib, and regorafenib, which inhibit several RTKs [49,58,59].

This multi-target action promotes a wide spectrum of side effects. Additionally, most patients develop resistance to these multitarget drugs by acquiring additional mutations. The development of RTK KIT allosteric modulators is a perspective and potent way to increase the selectivity and specificity of inhibitors.

The concept of designing conformational control inhibitors to target the activated form of kinases and enable the inhibition of a wide range of kinase mutants has been proposed. A new class of inhibitors targeting the 'switch control pocket' (allosteric inhibitors) has been reported [60,61]. Two of them, ripretinib and avapritinib, approved by the FDA in 2020, bind to the switch pocket through the activation loop and thus prevent the kinase domain from adopting an active state [62]. The analogue of these drugs, inhibitor 3G8 co-

crystallised with KIT (PDB ID: 6HH1), shows its location in the ATP-binding area adjacent to the αC-helix, similar to the ripretinib position [61]. This pocket was previously described in the comparative analysis of the crystallographic structure 1T45 and MD conformations of the inactive and constitutively active KIT states in which the JMR, KID, and C-tail regions are undefined or only partially defined [26].

We expect the comprehensive dynamic model of the full-length KIT cytoplasmic domain to significantly expand the search for new pockets. We used our long-scale unbiased MD simulations of the two proteins, KIT$^{WT}$ or KIT$^{D816V}$, to explore and characterise their pockets, and provide the best perspective for the development of selective allosteric modulators. In addition to detecting pockets, this search will also evaluate the mutation's impact on their location.

The localisation and characterisation of the pockets were carried out as follows: The search for optimal criteria for hunting pockets by testing different isovalues ranging from 0 to 1.0 in increments of 0.5 for both proteins results in two isovalues, 0.35 and 0.50, giving the maximum number of pockets in KIT$^{D816V}$ and KIT$^{WT}$, respectively (Figure S5). These two isovalues were used for subsequent pocket identification with the Fpocket protein cavity detection algorithm, which uses Voronoi tessellation, alpha shapes [63], and tracking of pocket volume changes along the concatenated trajectories of each protein. Finally, all the pockets were ranked according to the calculated volume as well as their local hydrophobic density.

A pocket search in KIT$^{WT}$ reveals a series of distinct pockets/cavities located within the TK domain and between the TK domain and the ID regions (Figure 9).

Since the shape and size that define the pocket volume, as well as the local hydrophobic densities of protein pockets and cavities, are important for binding site characterisation and structure-based drug design, the evolution of these metrics was recorded over the simulation time for each protein using the two isovalues (Figure S6). All pockets show a broad time-dependent volume distribution of binding sites, apparently due to the high plasticity of proteins.

The extended and voluminous top-ranked pocket **P1**, estimated with the isovalue of 0.35, shows a tunnel-like topology composed of the three interconnected pockets, which, with the isovalue of 0.50, dissociates into three separated pockets—the larger **P1**, and the smaller **P1′** and **P1″**. The pocket **P1** corresponds to the ATP-binding site that binds imatinib and other type 2 antagonists [22,23,38,49,64]. It represents a 'path-of-pockets 'with an unusually large binding volume, which is likely related to high adaptability of KIT's active site upon receptor activation and ATP binding. Generally, the flexibility of the active site leads to rearrangements of binding site groups, which are dependent on different ligands. For instance, next-generation drugs that have improved selectivity and reduced side effects, and are also effective in combating mutation-induced resistance, are also located in **P1** pocket but close to the αC-helix [61], as documented by the structure of KIT co-crystalised with the 3G8 inhibitor (PDB ID: 6HH1).

To validate the accuracy of the **P1** volume definition in MD conformations, we analysed the structures of RTK KIT co-crystalised with different ligands. In the ligand-removed KIT in the inactive state, the volume of the pocket accommodated type 2 inhibitors ranging from 1178 to 1553 Å$^3$ (Table S2). These values correspond well to those observed in the MD conformations of the inactive KIT$^{WT}$ with the isovalue of 0.35.

To the best of our knowledge, the other pockets found in KIT$^{WT}$, including the disconnected **P1′** and **P1″** from the **P1** pocket, have not been described in the literature and represent new allosteric pockets. The second pocket, **P2**, is formed by αC-helix, A-loop, and JMR residues. This pocket retains its integrity when using two isovalues (0.35 and 0.50) and only changes in volume, which differs by a factor of two. The third pocket, **P3**, showing a tunnel-like topology, is localised between KID, C-tail, and C-lobe, and can be either extended (isovalue of 0.35) with a volume comparable to **P2** or subdivided into two smaller pockets, **P3** and **P3′** (isovalue of 0.50). Neither of these two pockets, **P2** and **P3**, were described in literature as allosteric sites of inactive KIT$^{WT}$, but were detected in some

extreme conformations obtained from the 4 Å displacement (Normal Mode Analysis) of the partial structure of constitutively active KIT^D816V [26].

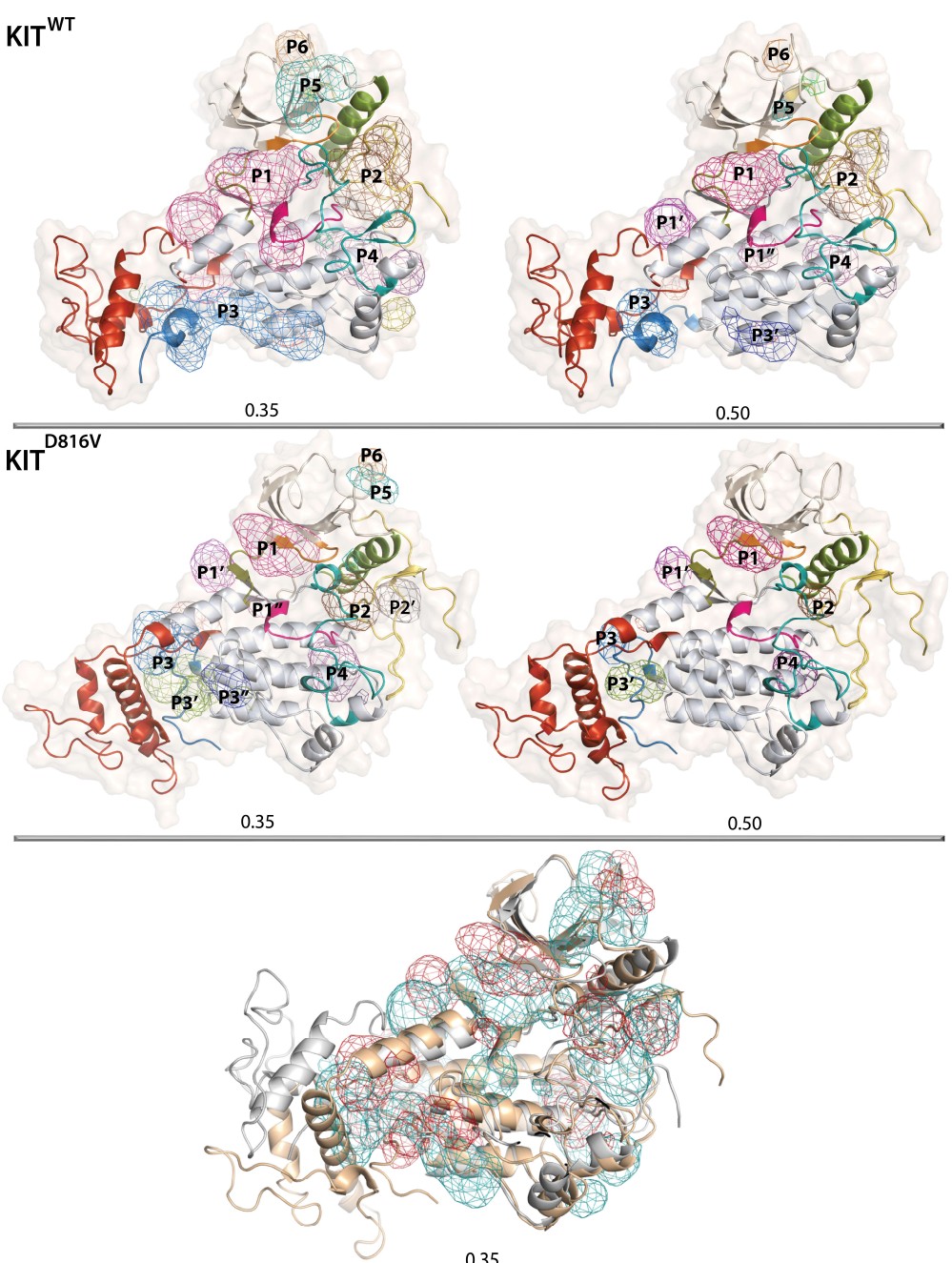

**Figure 9.** The RTK KIT POCKETOME. Searching of pockets and cavities in KIT^WT (**top**) and KIT^D816V (**middle**). Pockets were localised by the Fpocket algorithm using two isovalues, 0.35 (**left**) and 0.50 (**right**). Proteins are shows as cartoon with coloured domains/regions: the TKD N- and C-lobes are in beige and grey, respectively, JMR in yellow, P-loop in orange, C-loop in rose, αC-helix in green, A-loop in teal, KID in red, and C-tail in blue. The pockets are distinguished by coloured meshes and numbered according to their ranking in KIT^WT with isovalue 0.35. Superimposition of pockets in KIT^WT and KIT^D816V (**bottom**). KIT^WT and KIT^D816V are shown as grey and beige cartoons with the pockets in teal and red, respectively.

The **P4** pocket, located in the lobe C close to A-loop, is a bi-cavity pocket, viewed as a single entity (isovalue of 0.35) or two adjacent pockets (isovalue of 0.35). The N-lobe also processes two pockets, **P5** and **P6**, forming cavities on the protein surface, with a volume

highly dependent on the isovalue. Neither of these pockets (**P4**–**P6**) were described in the literature as allosteric sites in inactive $KIT^{WT}$.

By focusing on $KIT^{D816V}$, we observed the same pockets, **P1**–**P6**, as in $KIT^{WT}$ but with significantly reduced volume. Other differences seen in $KIT^{D816V}$ relate to a more fragmented **P3** pocket profile, which is subdivided into three separated pockets, one of which, **P3′**, is larger than **P1**, and the significantly decreased volume of **P5** and **P6**, which are more like small surface cavities than the pockets. The different location of the pockets and their sizes in the two KIT proteins illustrate that the binding site depends on the molecular conformations.

This search for KIT pockets with the use of two different isovalues and the comparison of the **P1** pocket with the literature and PDB data [65] suggests that the use of the 0.35 isovalue produced more suitable results.

To examine whether the newly identified pockets can be useful as the a priori appropriate targets of KIT proteins for the development of allosteric inhibitors, we compared their volumes with those validated in other proteins. For example, the analysis of HIV-1 proteases complexed with different inhibitors shows that the binding sites have a volume ranging from 540 to 875 $Å^3$ [66]. Consequently, the majority of the localised pockets have a satisfactory volume to accommodate a small molecule.

Both theoretical and experimental studies have shown that hydrophobic regions are the main contributors to pocket-binding affinity, while hydrophilic residues mainly contribute to drug specificity [67]. To better characterise the pockets located in the cytoplasmic domain of KIT, their hydrophobic scores were calculated. Although this parameter was calculated for two values (0.35 and 0.50), we only used the data obtained for 0.35, based on the observations made by the pocket volume estimation. According to the criteria defined in [68], the hydrophobic score (HS) value indicates the hydrophilic/hydrophobic properties of pockets. Scores ranged from −55 to −14 related to the hydrophilic surface of the pocket; scores from −10 to +13 are neutral, from +41 to +63 are hydrophobic, and from +74 to +100 are very hydrophobic.

As the hydrophobicity scores of KIT pockets range from −40 to +80, showing single or multiple probabilistic maxima, they exhibit all kinds of hydrophilicity/hydrophobicity with different probability (Figure S6). Pockets with hydrophobic or very hydrophilic properties are very rare. Most pockets exhibit neutral or moderate hydrophobic properties. In particular, a bimodal distribution of the **P1** pocket with two maxima, at 0 and 25, shows two kinds of hydrophobicity of the **P1** in $KIT^{WT}$ with different probability, the neutral (less probable) and hydrophobic (more probable). In the mutant, the ratio of neutral and hydrophobic **P1** was reversed. The monomodal distribution of **P2** with a maximum at HS of 20 in $KIT^{WT}$ and $KIT^{D816V}$ indicates that its equal hydrophobicity varied between neutral and moderately hydrophobic in both proteins. The HS probability of the **P3** pocket shows a unimodal distribution with $HS_{max}$ of 30 and 20 in $KIT^{WT}$ and $KIT^{D816V}$, respectively. The probability of the HS of the pocket **P4** has different $HS_{max}$, from 20 ($KIT^{WT}$) to 0 ($KIT^{D816V}$), both characterising neutral pockets. The small surface pockets, **P5** and **P6**, show a very similar HS distribution in both proteins describing equilibrated hydrophilicity/hydrophobicity. These particularities of each pocket composition are illustrated in Figure 10.

The pockets described here are, unfortunately, not an exhaustive list. We have observed additional pockets, such as those in KID and JMR. However, the algorithm used (Fpocket) does not allow us to fully characterize these pockets due to the limitation of the criteria used by PocketPicker to distinguish correctly identified binding sites from others [63].

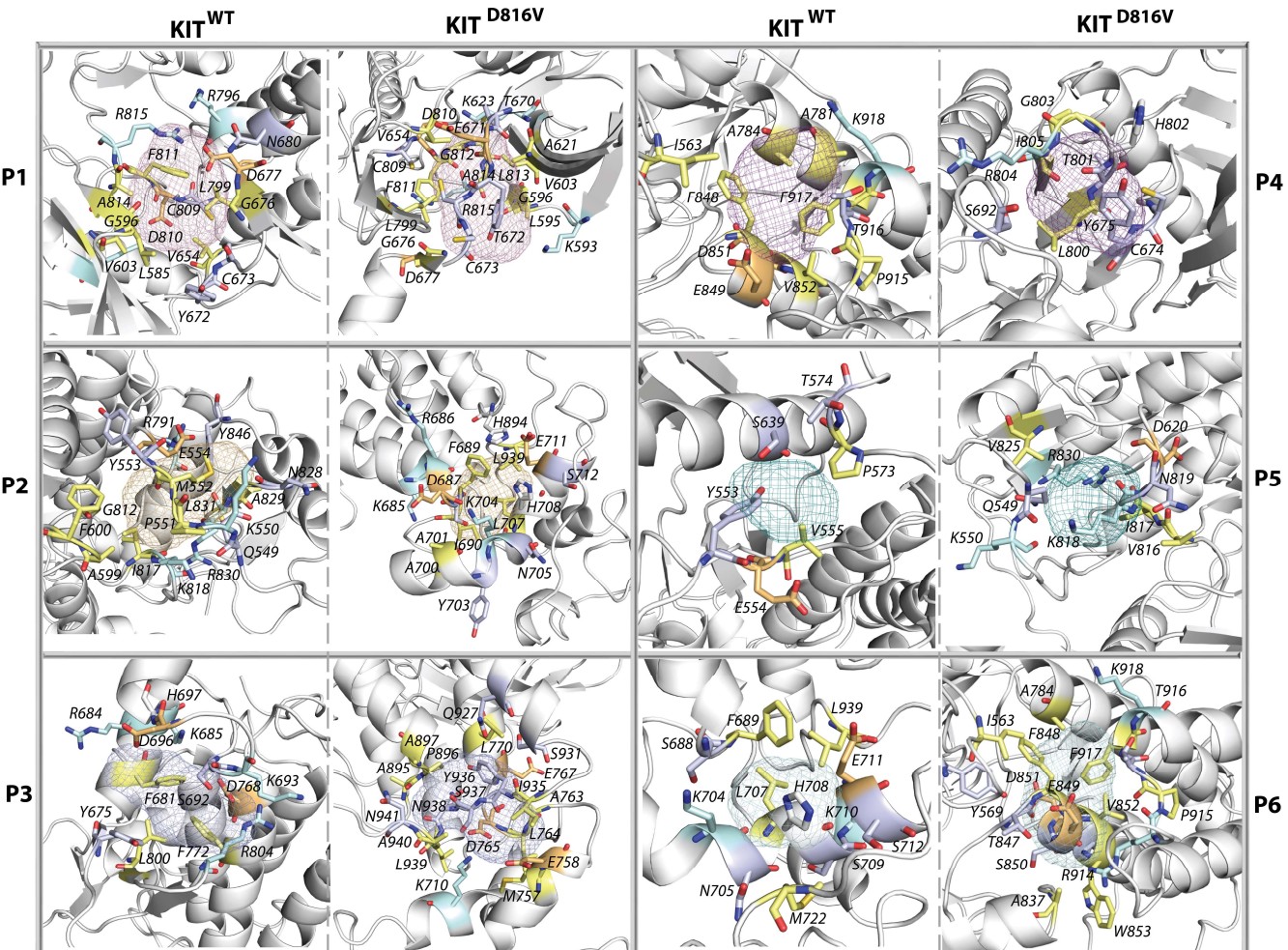

**Figure 10.** Surface of each pocket in KIT[WT] (**left**) and KIT[D816V] (**right**). Proteins are shown as cartoon. Distinct types of residues are distinguished by coloured sticks: polar residues are in light blue, hydrophobic are in pale yellow, positively charged are in pale cyan, and negatively charged are in light orange. For simplicity, the pockets are presented as defined with an isovalue of 0.50.

## 3. Discussion

The description of KIT[D816V] and KIT[WT] at the two fundamental levels, the inherent dynamics, and the dynamics of intra-molecular pockets, qualified respectively as DYNA-SOME [44] and POCKETOME [45], provided a crucial basis for the comparative analysis of two proteins differing only by the point mutation D816V.

DYNASOME-related analysis showed that both proteins are intrinsically disordered entities and consist of the virtually stable tyrosine kinase domain and four intrinsically disordered regions—JMR, KID, A-loop, and C-tail. DSSP assignment and clustering showed that the reversible secondary structure's transition from folded to alternatively folded or unfolded ($\alpha$H $\leftrightarrow$ 3$_{10}$-helix $\leftrightarrow$ $\beta$-strand $\leftrightarrow$ turn $\leftrightarrow$ bend $\leftrightarrow$ coil) is observed in each ID region of both proteins. The overall folding content is rather similar, except for A-loop, which exhibits the increased fold in KIT[D816V] compared to KIT[WT]. On the other hand, the estimation of the difference in probability of the secondary structure's formation for each pair of residues *i* of KIT[WT] and KIT[D816V] showed that the effects caused by the D816V mutation on KIT folding are also observed in the other regions, largely distant from the point mutation site, and they are specifically more pronounced in IDRs.

The dynamical conformational spaces of two proteins estimated by superimposition of the MD conformations into the two first principal components (PCA) show a significant overlap, suggesting, as a first approximation, a proximity of their conformational subsets.

The collective motions described with the first two PCA modes are strongly coupled in each KIT multidomain and between the domains. Global motions of the TK domain, specified as a pendulum-like circular movement along a virtual axis of rotation coinciding with the active site of each KIT, reveal the more pronounced and organised circularity in KIT$^{WT}$ compared to KIT$^{D816V}$. The intrinsic dynamics correlations, analysed with the calculated Pearson cross-correlation matrix for the all Cα-atom pairs of each protein, demonstrate strongly coupled movements within each KIT domain and between structural domains, even widely separated in 3D space. However, the cross-correlation pattern in KIT$^{D816V}$ is much more contrasting compared to KIT$^{WT}$ and therefore reflects a more coupled motion. The difference is mainly observed in the TK domain, in which the cross-correlations within and between the lobes are not identical in KIT$^{D816V}$ and KIT$^{WT}$. The more homogeneous block of the N-lobe in KIT$^{D816V}$ reflects the highly collective motions of the P-loop and αC-helix with JMR. Similarly, both segments of A-loop, flexible and buried, and the αC-helix show enhanced collective motions in KIT$^{D816V}$. This correlation of catalytic regions directly involved in KIT activation implies that point mutation affects their interactions by the motions altering.

Since coupling of protein movement reflects both short and long-distance allosteric regulation, enhancement of such coupling has functionally related content, indicating that KIT activation is more efficient in KIT$^{D816V}$. Earlier, we identified that the D816V mutation induces both short and long distance effects, promoting structural change in A-loop and JMR, and disruption of communication between these regions, maintained in KIT$^{WT}$ by H-bonds between the residues Y823 (A-loop) and E792 (C-loop) [25,69].

In the present study, we showed that the H-bond patterns stabilising the active site of KIT$^{D816V}$ are also significantly altered compared to KIT$^{WT}$. Moreover, in the multifunctional KIT, the D816V mutation also affects the distant JMR and KID, favouring their greater conformational diversity and promoting the dispersed location of tyrosine residues that can be responsible for the alternative recognition of the first-line (direct) interacting partners of KIT leading to the disorganized signalling networks. Interrelations between mutation-induced structural dynamics and alternation of protein activity are described for different proteins [70–73].

Intrinsic disorder is now considered a means to maximize allosteric binding in many protein families [74,75]. The idea is that the absence of a single 'rigid' structure confers a plasticity on the protein that readily supports dynamic changes in response to changes in its local environment, including binding to proteins or ligands. Therefore, the observed increase in conformational diversity of each ID region and the significant enhancement of dynamic coupling in the KIT$^{D816V}$ mutant result in the maximisation of mutant-induced allosteric regulation.

Assessing the overall plasticity of multidomain proteins containing disordered regions is significantly sophisticated compared to single-domain or well-ordered proteins. Using an explicit 'free energy landscape' model can illuminate different states of intrinsically disordered proteins. The relative Gibbs free energy, ΔG, defined on chosen coordinates, which are the 'collective variables', measures the probability of finding the protein of interest in two or more states and describes the intermediates between these states observed on the disordered sampling protein. Although our main objective is the evaluation of the effects induced by the point mutation, we suggested that the cumulative representation of two conformational spaces as a generic model of 'free energy landscape' could provide significant insights onto the main differences/similarities between these two proteins, KIT$^{D816V}$ and KIT$^{WT}$. These proteins differ only by the point mutation D816V, which can be considered as an effector promoting irreversible structural and conformational events in one of them.

As inherent motion is closely related to intermolecular interactions stabilising protein structure and conformation, the first principle components, PC1 and PC2 (PCA), describing the most essential collective motion were used as reaction coordinates for the evaluation relative free energy and the reconstruction of its landscape.

We observed that using PC1 and PC2 as reaction coordinates is apparently not an optimised choice for reconstructing 2D and 3D models of a generic 'free energy landscape' from the assembled conformational spaces of two disordered proteins. The cloud-like scatter plot of the multidomain proteins KIT$^{WT}$ and KIT$^{D816V}$ is derived from overlapping effects, which are either specific for each protein or common to both proteins. On the one hand, (i) the TK domain of two proteins shows the proximity of structural and conformational properties, and (ii) the intrinsically disordered regions suffer from the large inherent conformational divergence in each protein, which is more crucial in KIT$^{D816V}$. Nevertheless, a non-negligible portion of the IDR conformations of two proteins shows partial structural and conformational similarity. On the other hand, the ordered and disordered KIT regions exhibit the extrinsic disorder, evidenced by their different relative positions in 3D structure. Therefore, this complexity poses challenges in the interpretation the free energy maps.

As the FELs were reconstructed on the principal components, they represent only the conformations reflecting the 'essential' dynamics of the protein or its domain/region. The first two PCs represent only larger spatial scale movements and describe only 60 and 20% of variance, respectively, so the energy landscape, defined on these reaction coordinates, loses movements on a smaller spatial scale (e.g., intermediate conformations).

One of the promising approaches to choosing archetypal metrics to use as the reaction coordinates is the deciphering of extrinsic disorder-related motion into its linear and rotational components and their correlation. The other promising technique is the multiparameter clustering [76], which we used for the analysis of KID from KIT$^{WT}$ [77].

Nevertheless, some rational observations can be summarized from the study of conformational landscapes reconstructed on the principal components (PCA).

Despite numerous wells encompassing the conformations of both KIT species and showing their partial overlap in terms of folding and disordered position of the fragments relative to the TK domain, their content shows the main population of a protein. Additionally, a few wells only contain exclusively KIT$^{WT}$ or KIT$^{D816V}$. All these observations indicate an obvious conformational shift. Therefore, we suggest that changes in the dynamic and internal movements of KIT$^{D816V}$ are the main consequences of the mutation allowing its constitutively activated state. These changes favour the shift of the spontaneous equilibrium between active and inactive conformations prompted by mutation, leading to loss or gain of function, which could be associated with disease states [78]. These observations clearly highlight the potential for pharmacotherapeutics approaches to modulate the inappropriate functionality of the mutated receptor.

It is possible that the structural plasticity of KIT leads to the formation of hidden or cryptic pockets [79–81], which are the engaging sites for drug discovery, in outside visible pockets in experimentally determined structures [82]. Targeting these cryptic pockets offers a number of exciting opportunities from a drug development perspective, including the possibility of developing modulators that reduce off-target effects and have absolute subtype/mutant specificity.

Currently, computational methods (mainly MD simulations) capable of capturing and characterising these cryptic sites have restricted reliability due to the relatively limited empirical methods that allow direct validation of these pockets. Moreover, a detailed structural characterisation of the intrinsically or extrinsically disordered regions that give rise to cryptic sites is also poorly accessible to experimental methods.

A pocket search applied to the MD conformations of KIT$^{WT}$ reveals a large pocket formed with three interlinked pockets having a tunnel-like topology. The opening of a pocket on the surface, creating an accessible 'path' to the catalytic site, is observed in KIT$^{WT}$ conformations with a displaced JMR, even though the A-loop remains in an inactive state. In the KIT$^{D816V}$, this pocket is moved to the P-loop and its size has decreased due to its separation into three distant pockets. Still important in KIT$^{D816V}$, this pocket is classically used for the development of the ATP-competitive inhibitors and allosteric modulators bound to its area close to the αC-helix and distant JMR.

Although examples of pure allosteric modulators for this area are not yet available, all clinically used small molecular weight RTK TK domain inhibitors are competitive at ATP sites to some extent [83]. The best clinical examples of allosteric RTK inhibitors targeting the extracellular domain are either antibodies [75] or small molecules [84,85]. Two less-characterised RTK targets are the juxtamembrane and transmembrane regions. However, an allosteric agonist, gambogic amide, which selectively binds to this region of the TrkA RTK, has been reported [86].

Our discovery of the other pockets, which are newly identified allosteric sites having reduced volume in $KIT^{D816V}$ compared to $KIT^{WT}$, but still accessible to accommodate a small molecule, represents fertile ground for the future development of next-generation drugs targeting KIT that have an improved selectivity and reduced side effects, and they are also effective in combating mutation-induced resistance.

There have been significant advances in understanding the mechanisms of constitutive activation of RTK, and resistance mechanisms are leading to the development of next-generation drugs that not only have increased selectivity and reduced side effects but are also able to combat resistance caused by mutations. KIT is an archetypical example of such targets, but, unlike other proteins, all its inhibitors are multi-targeted. Nevertheless, there are significant opportunities for allosteric targeting of each receptor, which are not yet explored for KIT. Such investigation is very important because allosteric drugs can show conformational and site selectivity, crucial pathophysiological sensitivity, and, therefore, greater safety.

## 4. Methods

### 4.1. Modelling

***The full-length cytoplasmic domain linked to the transmembrane helix of $KIT^{D816V}$ mutant***. The 3D structure of the inactive $KIT^{WT}$ (L521-R946) was taken at 2 μs of the Molecular Dynamics (MD) simulation reported in [43]. Five thousand (5000) models of the full-length cytoplasmic domain of $KIT^{D816V}$ (L521-R946) having missense mutation D816V were generated for the human sequence P10721 (https://www.uniprot.org/uniprot/, accessed on 14 September 2020) with Modeller 10.1 [87] using the available structural data. The best model was chosen according to its DOPE score [88] and its stereochemical quality (Procheck) [89]. Further, the protein construct will be called $KIT^{D816V}$.

***The full-length cytoplasmic domain of KIT with transmembrane helix inserted into membrane.*** A phosphatidylcholine (POPC) lipid bilayer was generated using Charmm-Gui [90]. As a single transmembrane, α-helix was not found in the *Orientation of Protein in Membrane* (OPM) database [91], but the orientation of the double-helix of PDGFR-β, a cousin of KIT, was considered. Suggesting that the single helix in the monomer of $KIT^{D816V}$ can have an alternative orientation, $KIT_{L521-R946}$ was oriented manually so that its transmembrane helix was positioned perpendicularly to the bilayer with polar residues next to the phospholipids' polar heads and apolar residues among their tails. Finally, to reduce the number of residues located in the extracellular area, the N-terminal extremity of $KIT^{D816V}$ was reduced to I516-R946 ($KIT_{I516-R946}$).

### 4.2. Molecular Dynamics Simulation

**System set-up.** $KIT_{I516-R946}$, wrapped in a 23 Å-width leaflet lipid bilayer of POPC (2), was solvated with TIP3P water model in a rectangular box with the PACKMOL-Memgen [92] and LEaP modules of AmberTools20 (http://ambermd.org/AmberTools.php, accessed on 14 September 2020) using the ff14SB all-atom force field [93] for protein and Lipid17 for membrane: (i) hydrogen atoms were added; (ii) protonation states of amino-acids at physiological pH were assigned, and the histidine residues were protonated on their ε-nitrogen atoms; (iii) no counter-ions were added, as the system is already of neutral charge. The systems, $KIT_{T544-R946}$ in the water solution (1), and $KIT_{I516-R946}$ wrapped in a 23 Å-width leaflet lipid bilayer of POPC (2), contained 71,980 atoms in total with 6873 atoms of protein and 64,647 atoms

of water (1), and 172,368 atoms in total with 6869 atoms of protein, 30,144 atoms of the lipid membrane, 123,030 atoms of water (2), and one Cl$^-$ ion.

**Minimisation, equilibration and data generation**. The systems were equilibrated using the Sander module of AmberTools20. For system (1), 30,000 minimization steps were performed: (i) 10,000 on the all-atom fixed protein to relax the water; (ii) 10,000 with fixed C$\alpha$ atoms to allow the relaxation of sidechains; and (iii) 10,000 without any constraints on the system. For (2), the positional constraints of various forces were applied and subsequently decreased at each minimisation/equilibration step to allow a smooth equilibration. The values were for: all protein atoms—10, 10, 2.5, 1, 0.5, 0.1; the phosphate atom of POPC—2.5, 2.5, 1, 0.5, 0.1 kcal/mol; the dihedral angle around the double bond of the oleoyl chain of POPC (restricted to 0°) and the dihedral angle formed by the glycerol carbon and the oleoyl ester oxygen atoms of POPC (restricted to 120°)—250, 250, 100, 50, 25 kcal/mol. System (2) was minimised during 5000 steps (2500 steps of steepest descent then 2500 steps of conjugated gradient).

For both systems, the following steps were executed. A 100 ps thermalisation step was performed, where the temperature (atoms velocity) was gradually increased from 0 to 310 K using the Berendsen thermostat [94]. Then, a 100 ps equilibration step with constant volume and a 100 ps equilibration step with constant pressure (1 bar) were performed. Periodic boundary conditions and isotropic position scaling were imposed with the Berendsen barostat [94]. For these two steps, temperature regulation was performed with Langevin dynamics with a friction coefficient $\gamma$ = 1. Finally, a 100 ps molecular dynamics was completed at 310 K (Langevin dynamics), at constant volume and constant pressure, with a hybrid Monte-Carlo barostat [95]. In (2), the membrane surface tension was set to 0 on the x–y plane. Lastly, a mini (100 ps) molecular dynamics simulation in the previous conditions was completed without any constraint on the system.

All equilibration steps and molecular dynamics simulations were carried out with an integration step of 2 fs. Non-bonded interactions were calculated with the Particle-Mesh Ewald summation (PME) with a cut-off of 10 Å, and bonds involving hydrogen atoms were constrained with SHAKE algorithm [96]. The initial velocities were reassigned according to the Maxwell–Boltzmann distribution, and the same parameters (simulation conditions) as the mini dynamics were applied. Coordinates were recorded every 1 ps. The system was simulated with AMBER18 (http://ambermd.org/AmberMD.php, accessed on 14 September 2020) using the PMED Cuda module running on the supercomputer JEAN ZAY at IDRIS (http://www.idris.fr/jean-zay/, accessed on 14 September 2020). For system (1), a unique short trajectory of 1 ns and, for system (2), three extended trajectories of 2 µs were performed.

### 4.3. Data Analysis

All standard analyses were performed using the CPPTRAJ 4.25.6 program [97] of AmberTools20. Analysis of MD conformations (every 10 ps) was realized after least-square fitting on residues of the TK domain (W582-S688, L769-S931) or on residue-truncated trajectories of each fragment to remove rigid-body motions.

(1) The RMSD and RMSF values and cross-correlations were calculated for the C$\alpha$-atoms using the initial full-length model or each separated domain/region (at $t$ = 0 µs) as a reference.

(2) Secondary structural propensities for all residues were calculated using the Define Secondary Structure of Proteins (DSSP) method [98]. The secondary structure types were assigned for residues based on backbone -NH and -CO atom positions. Secondary structures were assigned every 10 and 20 ps for the individual and concatenated trajectories, respectively.

(3) Difference in the probability of formation of secondary structures for each pair of residues $i$ from KIT$^{WT}$ and KIT$^{D816V}$ was estimated as:

$$|^{WT}Phelix - {}^{MU}Phelix|$$

(1)

(4)   Clustering analysis was performed on the productive simulation time of each MD trajectory using an ensemble-based approach [55]. The algorithm extracts representative MD conformations from a trajectory by clustering the recorded snapshots according to their C$\alpha$-atom RMSDs. The procedure for each trajectory can be described as follows: (i) a reference structure is randomly chosen in the MD conformational ensemble, and all conformations within an arbitrary cut-off r are removed from the ensemble; this step is repeated until no conformation remains in the ensemble, providing a set of reference structures at a distance of at least r; (ii) the MD conformations are grouped into n reference clusters based on their RMSDs from each reference structure. The cut-off was varied from 3 to 5 Å. The analysis was performed every 100 ps.

(5)   The H-bonds between donor (D) and acceptor (A) atoms N, O, S were monitored according to the following geometrical parameters: d(D-A) $\leq$ 3.6 Å, $\angle$(DHA) $\geq$ 120°. Hydrophobic contacts were considered for all hydrophobic residues with side chains within a 4 Å of each other.

(6)   The Principal Components Analysis (PCA) modes were calculated for the backbone atoms (N, H, C$\alpha$, C, O) after least-square fitting on the average conformation calculated on the concatenated data. The eigenvectors were visualized with NMWiz module for VMD [99].

(7)   The relative Gibbs free energy of the canonical ensemble was computed as a function of two reaction coordinates with Equation (2) [100]:

$$\Delta G = -k_B T \, ln \, \frac{P(R_1, R_2)}{P_{max}(R_1, R_2)} \tag{2}$$

where $k_B$ represents the Boltzmann constant, and $T$ is the temperature. $P(R_1, R_2)$ denotes the probability of states along the two reaction coordinates, which is calculated using their joint probability, and $P_{max}(R_1, R_2)$ denotes the maximum probability. The population of each well was roughly estimated using a square defined with $R_1$ and $R_2$ value intervals and containing red to orange $\Delta G$ colors.

(8)   The pocket prediction protocol includes three steps: (i) Finding optimal criteria for pocket hunting. This step was performed by testing different isovalues ranging from 0 to 1.0 in 0.5 increments for both proteins. Two isovalues, 0.35 and 0.50, give the maximum number of pockets in KITD816V and KITWT, respectively. (ii) Pockets were identified using by Fpocket protein cavity detection algorithm, which uses Voronoi tessellation and alpha shapes [63]. (iii) Tracking the change in pocket volume along the concatenated trajectories of each protein was performed using two isovalues, 0.35 and 0.50. (iv) Pockets were ranked based on the calculated volume as well as on their local hydrophobic density.

*4.4. Visualisation and Figure Preparation*

Visual inspection of the conformations and figure preparation were performed with PyMOL (https://pymol.org/2/, accessed on 14 September 2020). The VMD 1.9.3 program (accessed on 21 December 2020) [101] was used to prepare the protein MD animations. To visualise the motions along the principal components, the Normal Mode Wizard (NMWiz) plugin [99,102] which is distributed with the VMD program, was used. The 3-dimensional representations of the free energy surface were plotted using Matlab (US, © 1994-2021 The MathWorks, Inc., Natick, MA, USA).

**Supplementary Materials:** The following are available online at https://www.mdpi.com/article/10.3390/kinasesphosphatases1040014/s1: Figure S1: Homology modelling of KIT[D816V] from the KIT[WT]; Figure S2: Molecular Dynamics (MD) simulations of the full-length cytoplasmic domain of KIT[D816V] and KIT[WT]; Figure S3: Folding of KIT[D816V] and KIT[WT]; Figure S4: PCA of the MD conformations of KIT; Figure S5: Search of the optimal criteria for pocket hunting; Figure S6: The RTK KIT POCKETOME characterisation; Table S1: Folding of the intrinsically disordered regions in KIT[WT] and KIT[D816V]; Table S2: Pocket characterisation in KIT[WT] and KIT[D816V].

**Author Contributions:** Conceptualization, L.T.; methodology, J.L., M.B. and L.T.; software, J.L. and M.B.; validation, J.L. and L.T.; formal analysis, J.L. and M.B.; investigation, J.L. and L.T.; resources, J.L. and L.T.; data curation, J.L.; writing—original draft preparation, L.T.; writing—review and editing, J.L. and L.T.; visualization, J.L., M.B. and L.T.; supervision, L.T.; project administration, L.T.; funding acquisition, L.T. All authors have read and agreed to the published version of the manuscript.

**Funding:** This research was funded by Ministère de l'Enseignement supérieur, de la Recherche et de l'Innovation, FRANCE (scholarship J.L.).

**Institutional Review Board Statement:** Not applicable.

**Informed Consent Statement:** Not applicable.

**Data Availability Statement:** The numerical model simulations upon which this study is based are too large to archive or to transfer. Instead, we provide all the information needed to replicate the simulations. The model coordinates are available from L. Tchertanov at ENS Paris-Saclay.

**Acknowledgments:** This research was supported by Centre National de la Recherche Scientifique (CNRS) and Ecole Normale Supérieure (ENS) Paris-Saclay. The authors were granted access to high-performance computing (HPC) resources at the French National Computing Centre CINES (DARI A0070710973) by GENCI (Grand Equipement National de Calcul Intensif). Calculations were performed on the Jean Zay cluster at IDRIS (101063).

**Conflicts of Interest:** The authors declare no conflict of interest. The funders had no role in the design of the study; in the collection, analyses, or interpretation of data; in the writing of the manuscript; or in the decision to publish the results.

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
