# Peer review of "Receptor Tyrosine Kinase KIT: Mutation-Induced Conformational Shift Promotes Alternative Allosteric Pockets"

_2813-3757, doi:10.3390/kinasesphosphatases1040014_

Round 1

Reviewer 1 Report

This is a highly specialized study presenting a thorough in silico analysis of the differences between KIT and its oncogenic mutant D816V KIT, with the commendable intent of disclosing allosteric pockets exploitable for the selective inhibition of the mutant.

Some minor points:

1.The D816V KIT mutant is often described in the manuscript as a constitutively active kinase. However, its description in the Introduction would instead suggest a decreased activity of the mutant compared to the WT KIT; see page 3, line 94-100. (“... did not dimerise... decreased tyrosine kinase domain stability...”). The catalytic features of the mutant as per the literature should be better explained.

2. The language requires a thorough check. Just some examples (but please, check all the manuscript, many other errors are present):

- in the title: alternating?

- lines 104-106: several errors (“represents a multitarget DRUG (WHICH inhibits specifically ABL, BCR-ABL, KIT, and PDGFRα) which IS widely used for treatment of many tumors [34]. However, in vitro investigations ON the efficacy of imatinib ...”

 - line 318 “Principle” should be “main”

- line 367: is differ... (and in several points “differed” often used instead of “different”)

- sentence of lines 428-431: a verb is missing

- line 452: “What ARE WE learning...”

- line 684: “To validate the accuracy correctness ...”

- line 702: “The N-lobe processes also two pockets....”

3. I suggest to do an effort to add a simplified explanation of the applied methods (in the Introduction or in the Results sections), for readers who are not familiar with computational biology. For example, at the end of the Introduction, where POCKETOME, DYNAZOME and INTERACTOME are mentioned, beside the references, maybe some explicative words would be useful.

The language requires a thorough check (see above)

Author Response

Reviewer 1

This is a highly specialized study presenting a thorough in silico analysis of the differences between KIT and its oncogenic mutant D816V KIT, with the commendable intent of disclosing allosteric pockets exploitable for the selective inhibition of the mutant.

Response: The Authors thank Reviewer 1 for the positive comments on the manuscript and critical remarks which were considered in the revised version of the manuscript. The manuscript was carefully checked for British English. As almost all corrections always correspond to syntax errors or typos, we do not explain each correction, but their position is easily visible in the text.

Some minor points:

1.The D816V KIT mutant is often described in the manuscript as a constitutively active kinase. However, its description in the Introduction would instead suggest a decreased activity of the mutant compared to the WT KIT; see page 3, line 94-100. (“... did not dimerise... decreased tyrosine kinase domain stability...”). The catalytic features of the mutant as per the literature should be better explained.

Response: In the introduction, we cited experimental results directly related to the themes of our study. We mentioned (i) the empirically proven monomeric state of the KIT mutant to justify our monomeric model, and (ii) the experimentally observed decrease in the stability of the tyrosine kinase domain, a point requiring careful study which was carried out as part of our work. All detailed explanations of the experimental results can be found in the recently published original article (Rajan et al., 2022). The catalytic characteristics of the mutant are not the subject of our study.

  1. The language requires a thorough check. Just some examples (but please, check all the manuscript, many other errors are present):

- in the title: alternating?

- lines 104-106: several errors (“represents a multitarget DRUG (WHICH inhibits specifically ABL, BCR-ABL, KIT, and PDGFRα) which IS widely used for treatment of many tumors [34]. However, in vitro investigations ON the efficacy of imatinib ...”

 - line 318 “Principle” should be “main”

- line 367: is differ... (and in several points “differed” often used instead of “different”)

- sentence of lines 428-431: a verb is missing

- line 452: “What ARE WE learning...”

- line 684: “To validate the accuracy correctness ...”

- line 702: “The N-lobe processes also two pockets....”

Response: We completely agree with the Referee’s recommendation to check the language. All error corrections mentioned and more have been corrected throughout the manuscript.

  1. I suggest to do an effort to add a simplified explanation of the applied methods (in the Introduction or in the Results sections), for readers who are not familiar with computational biology. For example, at the end of the Introduction, where POCKETOME, DYNAZOME and INTERACTOME are mentioned, beside the references, maybe some explicative words would be useful.

Response: Each of the highlighted words (DYNASOME, POCKETOME and INTERACTOME) in the manuscript is preceded by its definition applied to KIT.

  • the inherent dynamics of KIT (DYNASOME [44]),
  • describing the allosteric pockets (POCKETOME [45])
  • interactions with protein partners (INTERACTOME [46])

We agree that readers unfamiliar with such concepts may have difficulty identifying such definitions in their respective sentences. We hope that our additions will clarify the definitions

“Thus, clarifying the inherent dynamics of KIT (DYNASOME [44]), describing the ensemble of its pockets (POCKETOME [45]) and identifying KIT’s interactions with all its protein partners (INTERACTOME [46]) could help develop highly effective KIT-specific inhibitors acting simultaneously on intramolecular targets and the inter-face between interacting proteins ̶ allo-network drugs [47].”

Reviewer 2 Report

The manuscript is quite well written. I suggest to improve sevarl parts such as:

1) Receptor tyrosine kinase (RTK) KIT is key regulator of cellular signalling and its deregulation promotes the development and progression of many sever diseases (e.g., cancers). Mutation-induced effects lead to constitutive activation of the KIT cytoplasmic domain that causes aberrant intracellular signalling. We report the first 3D dynamical model (DYNAZOME) of the full-length cytoplasmic domain of the oncogenic mutant KITD816V generated by long-timescale unbiased MD simulations. Comparison of the structural and dynamical properties of KITD816V with those of wild-type KIT (KITWT) allowed evaluation of impact induced by mutation on each protein domain, including well-ordered and intrinsically disordered functional regions. Both proteins were compared in terms of the free-energy landscape and intramolecular coupling. The pockets search detected the new allosteric pockets in each protein, KITD816V and KITWT (POCKETOME). These pockets open an avenue for the development of novel highly selective KITD816V-specific allosteric modulators.

Please improve the description of the abstract and clarify the aim, results and conclusions.

2) 1. Introduction 25 Receptor tyrosine kinases (RTKs) are the canonical membrane proteins that control 26 the signal transduction of extracellular signals to the nucleus through tightly coupled 27 signalling cascades which alter the expression pattern of numerous genes [1-3]. RTK KIT, 28 also known as the CD117 differentiation cluster, is a RTK family III consisting of 976 amino 29 acids (aas). Its activity is regulated by a highly specific cytokine, the Stem-Cell Factor 30 (SCF), acting as messenger to initiate signal transduction. Stimulation by SCF in the 31 extracellular medium enables KIT to recruit protein partners through the cytoplasmic 32 domain. KIT initiates critical signalling pathways through multiple specific 33 phosphotyrosines which bind the downstream proteins containing Src homology (SH2) 34 or phosphotyrosine-binding (PTB) domains, and propagates the SCF-induced signal to 35 nuclei [4-6], Improve the introduction and underline the novelty of the study.

3) To the best of our knowledge, we report for the first time an exhaustive comparison of an inactive state of KITWT 149 and one of its oncogenic constitutively active mutants using the 150 most complete and advanced 3D dynamical models. Improve the description of paper aim and underline the most important aims of the study. 

4) 2. Results 152 2.1. Data Generation and Proceeding. Please underline the most important results to clarify all the results section.

5) 3. Discussions The description of KITD816V and KITWT 758 at the two fundamental levels ̶ the inherent 759 dynamics and the dynamics of intra-molecular pockets, qualified respectively as 760 DYNAZOME [44] and POCKETOME [45], provided a crucial basis for the comparative 761 analysis of two proteins differing only by the point mutation, D816V. Please, summarise here the most important results.

The manuscript is quite well written.  I suggest a minor revision of English language.

Author Response

Reviewer 2

The Article entitled “The Inherent Coupling of Intrinsically Disordered Regions in the Multidomain Receptor Tyrosine Kinase KIT”, submitted by Ledoux J. et al., reports the investigation on the role of the KIT cytoplasmic domain in triggering several signaling cascades. In particular, authors propose a 3D model to describe KIT intrinsic and extrinsic functionality, suggesting that all properties of KIT in the active status depend on events happening in the inactive status. In my opinion, this is a complete and well done study. However, if discussion, methods and data analysis clearly describe the project, results are confusing in some parts, and should be revised in the explanation. Lastly, I recommend English revision and editing.

Response: The Authors thank Reviewer 2 for very positive comments on the manuscript and critical remarks which were considered in the revised version of the manuscript. Our responses are supplied after each minor concern. The manuscript was carefully checked for British English. As almost all corrections always correspond to syntax errors or typos, we do not explain each correction, but their position is easily visible in the text.

The manuscript is quite well written. I suggest to improve sevarl parts such as:

  • Receptor tyrosine kinase (RTK) KIT is key regulator of cellular signalling and its deregulation promotes the development and progression of many sever diseases (e.g., cancers). Mutation-induced effects lead to constitutive activation of the KIT cytoplasmic domain that causes aberrant intracellular signalling. We report the first 3D dynamical model (DYNAZOME) of the full-length cytoplasmic domain of the oncogenic mutant KITD816V generated by long-timescale unbiased MD simulations. Comparison of the structural and dynamical properties of KITD816V with those of wild-type KIT (KITWT) allowed evaluation of impact induced by mutation on each protein domain, including well-ordered and intrinsically disordered functional regions. Both proteins were compared in terms of the free-energy landscape and intramolecular coupling. The pockets search detected the new allosteric pockets in each protein, KITD816V and KITWT (POCKETOME). These pockets open an avenue for the development of novel highly selective KITD816V-specific allosteric modulators.

Please improve the description of the abstract and clarify the aim, results and conclusions.

Response: The Authors thank Reviewer 2 for very positive comments on the manuscript and critical remarks which were considered in the revised version. Our responses are supplied after each minor concern.

ABSTRACT (New version)

Receptor tyrosine kinase (RTK) KIT is a key regulator of cellular signalling, and its dysregulation contributes to the development and progression of many serious diseases. Several mutations lead to the constitutive activation of the cytoplasmic domain of KIT, causing the aberrant intracellular signalling observed in malignant tumours. Elucidating the molecular basis of mutation-induced effects at the atomistic level is absolutely required. We report the first dynamic 3D model (DYNASOME) of the full-length cytoplasmic domain of the oncogenic mutant KITD816V, generated through unbiased long-scale MD simulations under conditions mimicking the natural environment of KIT. The comparison of the structural and dynamical properties of multidomain KITD816V with those of wild-type KIT (KITWT) allowed us to evaluate the impact of the D816V mutation on each protein domain, including multifunctional well-ordered and intrinsically disordered (ID) regions. The two proteins were compared in terms of free energy landscapes and intramolecular couplings. The increased intrinsic disorder and gain of coupling within each domain and between distant domains in KITD816V demonstrate its inherent self-regulated constitutive activation. The search for pocket revealed novel allosteric pockets (POCKETOME) in each protein, KITD816V and KITWT. These pockets open an avenue for the development of new highly selective allosteric modulators specific to KITD816V.

  • Introduction

 Receptor tyrosine kinases (RTKs) are the canonical membrane proteins that control the signal transduction of extracellular signals to the nucleus through tightly coupled  signalling cascades which alter the expression pattern of numerous genes [1-3]. RTK KIT, also known as the CD117 differentiation cluster, is a RTK family III consisting of 976 amino acids (aas). Its activity is regulated by a highly specific cytokine, the Stem-Cell Factor  (SCF), acting as messenger to initiate signal transduction. Stimulation by SCF in the  extracellular medium enables KIT to recruit protein partners through the cytoplasmic  domain. KIT initiates critical signalling pathways through multiple specific phosphotyrosines which bind the downstream proteins containing Src homology (SH2) or phosphotyrosine-binding (PTB) domains, and propagates the SCF-induced signal to nuclei [4-6].

Improve the introduction and underline the novelty of the study.

Response:  The citation refers to the introduction of RTK KIT and its functional role. The novelty of the study is reported in the last two paragraphs of the introduction.

  • To the best of our knowledge, we report for the first time an exhaustive comparison of an inactive state of KITWT and one of its oncogenic constitutively active mutants using the most complete and advanced 3D dynamical models.

Improve the description of paper aim and underline the most important aims of the study. 

Response:  The aims of our study are formulated as a series questions (see lines 122-148) which were studied for the first time (149-151).

  • Results 2.1. Data Generation and Proceeding. Please underline the most important results to clarify all the results section.

Response:  The “data generation and processing” is not a “result” per se but a summary on how the data were obtained pre-processed before all following analysis. It must be done correctly and therefore described. Each step of analysis produces results that are all important and clearly explained in the text. Each result opens the next step of analysis. All steps are logically interconnected. Conclusions can only be drawn from all the results together. This is theoretical work (numerical semi-empirical simulations, analysed by mathematical methods) and not experimental observations carried out using various methods. In our work, we move from a simple description of the system to a more complex one and further trace the coherence between all levels of description.

  • Discussions The description of KITD816V and KITWT at the two fundamental levels ̶ the inherent  dynamics and the dynamics of intra-molecular pockets, qualified respectively as  DYNAZOME [44] and POCKETOME [45], provided a crucial basis for the comparative analysis of two proteins differing only by the point mutation, D816V.

Please, summarise here the most important results.

Response:  I can't name the most "important" results - they were all obtained for the first time, so they are all important. Importance is a relative concept and is mainly determined by the reader's profile. In this work, the results obtained are addressed both to developers of new methods for analysing such complex proteins, as well as researchers who study such systems, pharamacologists, chemists... I don't know what is important for each category of researchers.

Round 2

Reviewer 2 Report

No further comments

 Minor changes of English language are required